# ER-associated degradation regulates Alzheimer's amyloid pathology and memory function by modulating γ-secretase activity

Bing Zhu[1], LuLin Jiang[1], Timothy Huang[1], Yingjun Zhao[1], Tongfei Liu[1], Yongwang Zhong[2], Xiaoguang Li[1], Alexandre Campos[3], Kenneth Pomeroy[3], Eliezer Masliah[4], Dongxian Zhang[1] & Huaxi Xu[1,5]

Endoplasmic-reticulum-associated degradation (ERAD) is an important protein quality control system which maintains protein homeostasis. Constituents of the ERAD complex and its role in neurodegeneration are not yet fully understood. Here, using proteomic and FRET analyses, we demonstrate that the ER protein membralin is an ERAD component, which mediates degradation of ER luminal and membrane substrates. Interestingly, we identify nicastrin, a key component of the γ-secretase complex, as a membralin binding protein and membralin-associated ERAD substrate. We demonstrate a reduction of membralin mRNA and protein levels in Alzheimer's disease (AD) brain, the latter of which inversely correlates with nicastrin abundance. Furthermore, membralin deficiency enhances γ-secretase activity and neuronal degeneration. In a mouse AD model, downregulating membralin results in β-amyloid pathology, neuronal death, and exacerbates synaptic/memory deficits. Our results identify membralin as an ERAD component and demonstrate a critical role for ERAD in AD pathogenesis.

[1] Neuroscience Initiative, Sanford Burnham Prebys Medical Discovery Institute, La Jolla, CA 92037, USA. [2] Center for Biomedical Engineering and Technology, University of Maryland School of Medicine, Baltimore, MD 21201, USA. [3] Proteomics Facility, Sanford Burnham Prebys Medical Discovery Institute, La Jolla, CA 92037, USA. [4] Departments of Neurosciences and Pathology, University of California, San Diego, La Jolla, CA 92093, USA. [5] Fujian Provincial Key Laboratory of Neurodegenerative Disease and Aging Research, Institute of Neuroscience, College of Medicine, Xiamen University, Xiamen 361005 Fujian, China. Bing Zhu and LuLin Jiang contributed equally to this work. Correspondence and requests for materials should be addressed to H.X. (email: xuh@sbpdiscovery.org)

Alzheimer's disease (AD) is characterized by β-amyloid (Aβ) plaques and neurofibrillary tangles (NFTs) as well as synaptic degeneration and memory/cognitive deficits. Aβ is generated by sequential cleavage of the transmembrane β-amyloid precursor protein (APP) by β-secretase and the γ-secretase complex. Growing evidence indicates that Aβ oligomers are neurotoxic, and trigger a cascade of pathological events culminating in eventual neurodegeneration[1]. In support of this, familial AD mutations which enhance amyloidogenic Aβ42 production have been identified in presenilin-1 (PS1) subunits of the γ-secretase complex, and APP. The γ-secretase complex is heteromeric protease complex comprising PS1, nicastrin, anterior pharynx-defective 1 (APH-1), and presenilin enhancer 2 (PEN2) subunits, where PS1 mediates proteolytic catalysis[2] and nicastrin functions as a substrate receptor within the complex and acts as a stabilizer[3]. Notch and APP are two critical γ-secretase substrates that are proteolytically processed into Notch-1 intracellular domain (NICD) and Aβ fragments, respectively[4,5].

The endoplasmic reticulum (ER) is a key cellular organelle required for protein synthesis and folding. The ER-associated degradation (ERAD) system enforces cellular protein quality control by translocating improperly-folded or unwanted proteins from the ER into the cytosol for proteasomal degradation[6–8]. Several ERAD components, including Derlin1[9], Erlin2[10], Ubiquilin2[11], VCP[12] and SYVN1[13] (also known as Hrd1), have been linked to multiple neurodegenerative diseases, including amyotrophic lateral sclerosis, and Parkinson's disease. It has been known that the ERAD system utilizes different sub-complexes to

degrade ER luminal (ERAD-L) and membrane (ERAD-M) substrates. Ubiquitin E3 ligases such as SYVN1 and AMFR (also known as GP78) play a central role in organizing these subnetworks[14]. For example, the SYVN1-centered subnetwork degrades both ERAD-L and ERAD-M substrates[15–17], while the AMFR subnetwork primarily degrades ERAD-M substrates[18].

Membralin (also known as TMEM259) is an evolutionarily conserved ER protein with several predicted transmembrane loops which lacks any domains[19]. Our work previously showed that a complete loss of membralin in mice leads to severe early-onset motor neuron degeneration, resulting in death around postnatal day 5; heterozygous membralin knockout mice exhibit no obvious adverse phenotype[20]. While ER-associated stress/unfolded protein responses (UPR) are associated with neuronal degeneration[21–23], how membralin is precisely linked to neurodegenerative disorders is yet unclear. A recent genome-wide association (GWAS) study has shown that the *membralin* (also known as *C19ORF6* in human) gene is located within 500 bp of a single nucleotide polymorphism (SNP) locus tightly associated with late-onset AD[24], and an additional study demonstrated that *membralin* transcript splicing is significantly altered in AD[25]. Using interactome network analysis, we identify and confirm herein that membralin is a component of the ERAD network to maintain homeostatic degradation of both luminal and membrane substrates, and pathophysiological substrates such as nicastrin. Membralin mRNA and protein levels are found to be markedly decreased in AD brain. Moreover, membralin deficiency increases γ-secretase activity, leading to elevated Aβ and

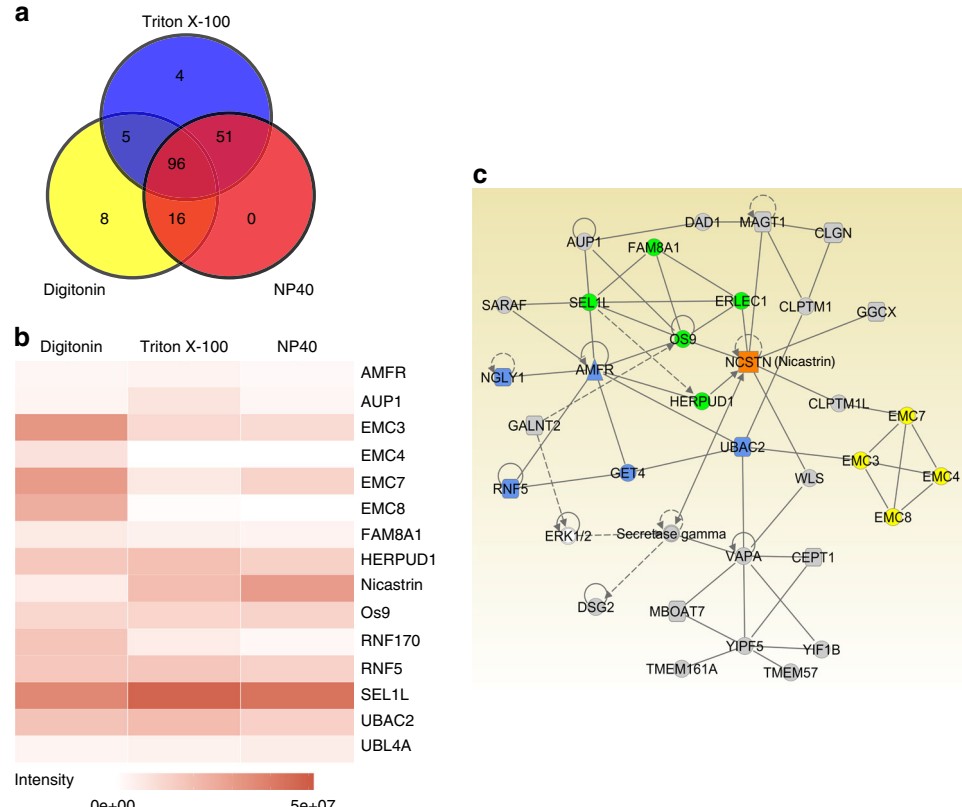

**Fig. 1** Proteomic analysis of membralin-interacting proteins. **a** Venn diagram showing the number of interacting proteins identified from cell lysates prepared in NP40 (*red*), Digitonin (*yellow*), and TritonX-100 (*blue*). **b** Among 180 proteins identified in the membralin interactome (see Supplementary Fig. 1), a heatmap of 14 known ERAD components is presented. The color intensity represents protein intensity (summed peptide area under the curve) (**c**) Interaction network of the membralin interactome by IPA. Three main ERAD subnetworks were highlighted in green (SYVN1 subnetwork), blue (AMFR subnetwork) and yellow (EMC subnetwork). Nicastrin was highlighted in orange color. Solid and dashed lines represent direct interactions and indirect interactions, respectively

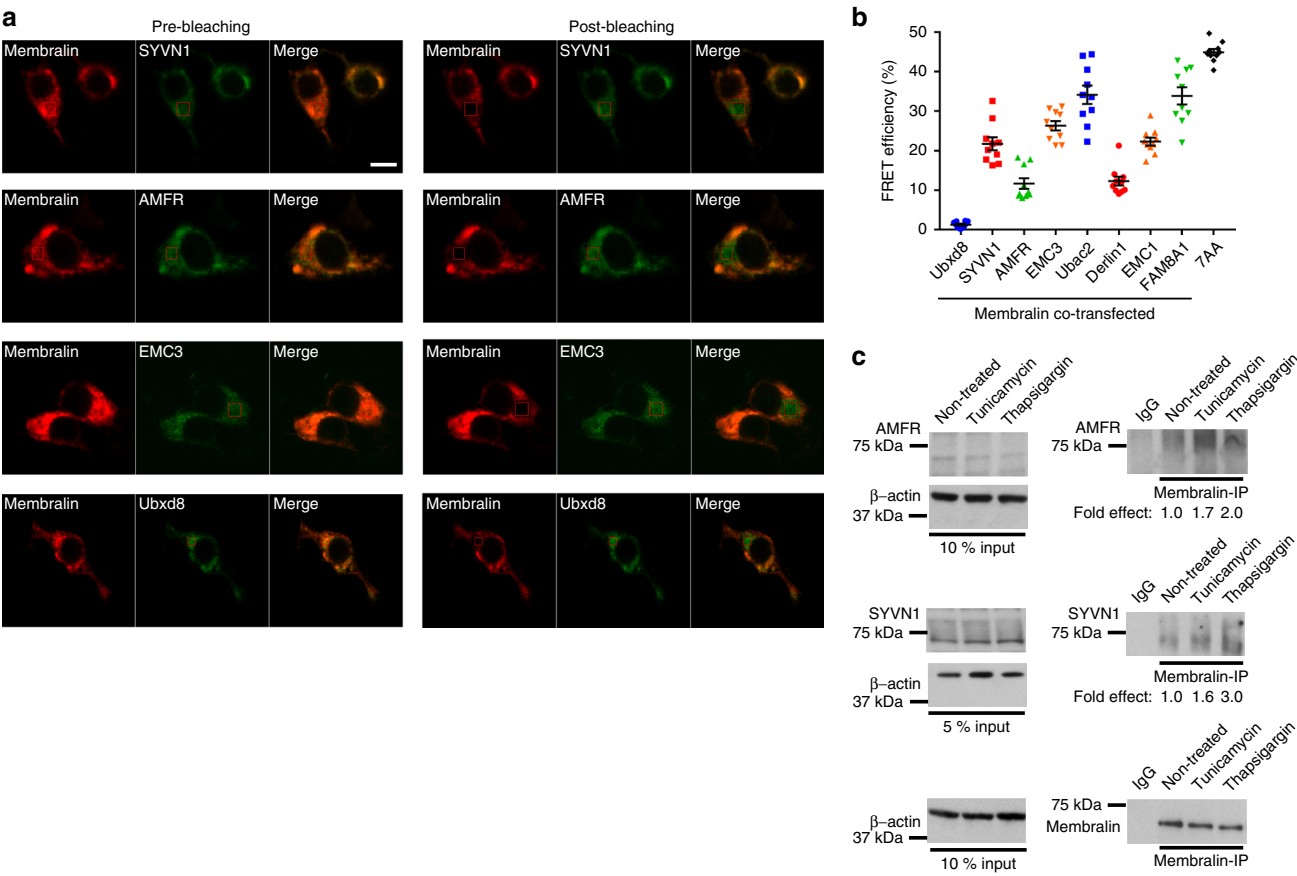

**Fig. 2** Confirmation of membralin-interacting protein components by FRET and co-IP. **a** FRET images of cells co-transfected with vectors expressing membralin-mCherry and either SYVN1-EGFP, AMFR-EGFP, EMC3-EGFP or Ubxd8-EGFP. Scale bar, 10 μm. **b** Quantification of FRET efficiency between membralin and SYVN1, AMFR, EMC3, Ubxd8, Derlin1, EMC1, Ubac2, FAM8A1, respectively. mCherry-7AA-EGFP was used as a positive control ($n = 10$ cells). Data represent mean ± s.e.m. from three independent experiments. **c** Co-IP analysis of endogenous membralin interactions with endogenous SYVN1, or AMFR in N2a cells under normal and ER stress conditions induced by tunicamycin (5 μg/ml) or thapsigargin (500 nM) for 4 hrs. The fold effect shown was calculated using the ratio from AMFR or SYVN1 (IP/Input) normalized to the membralin IP (IP/β-actin). Non-treated cell samples were set to 1.0

NICD generation. Downregulating membralin results in Aβ plaque pathology, neuronal death and marked exacerbation of synaptic and memory deficits in a mouse model of AD. These results demonstrate a novel role for membralin-associated ERAD function in amyloidogenic AD pathogenesis.

## Results

**Identification of membralin as a key ERAD component.** In order to elucidate a potential role for membralin in the ER, we searched for membralin-interacting proteins by immunoprecipitating Myc-tagged mouse membralin complexes from HEK293T cells and characterizing bound components by affinity-purification mass spectrometry (AP-MS)[14]. To probe the integrity of protein complexes under various detergents, we individually immuno-purified membralin complexes from lysis buffers containing Digitonin, Triton X-100, and Nonidet P-40 (NP40), respectively. Using stringent filtering criteria for positive inter-actors, we identified 180 potential membralin-interacting proteins (Supplementary Fig. 1 and Supplementary Table 1). Of the 180 proteins identified, 96 proteins components co-precipitated with membralin under all three detergent conditions: very few of the components identified without overlapping between the three extraction conditions, indicating comparable integrity of the membralin complex in the presence of various detergents (Fig. 1a). Using Ingenuity Pathway Analysis (IPA), we characterized the membralin-interaction network, which notably

comprised numerous ERAD interacting molecules (Fig. 1b, c). Within this network, we observed membralin interactions with three known ERAD subnetworks[14]; namely AMFR- (AMFR, GET4, Ubac2, NGLY1, RNF5), SYVN1- (ERLEC1, FAM8A1, SEL1L, OS9, HERPUD1), and EMC- (EMC3, EMC4, EMC7, EMC8) subnetworks, respectively. Identification of ERAD components in the membralin interactome and its association with functional ER pathways indicate that membralin may be an ERAD component and is potentially involved in multiple ER-associated processes. To validate potential protein interactions obtained from proteomic analysis, we determined whether mCherry-tagged membralin interacted with GFP-tagged constructs comprising four representative ERAD proteins (EMC3, Ubac2, FAM8A1, and AMFR) obtained from the proteomic dataset. Individual co-expression of all four proteins in HEK293T cells showed clear co-localization with membralin as visualized by confocal microscopy (Supplementary Fig. 2a–d). Using this assay, we tested co-localization of membralin with six additional key ERAD components, including SYVN1 (E3 ligase), EMC1 (EMC component), Derlin1 (dislocation protein), Ubxd8, VCP and Ubiquilin2 (proteasome associated protein). We found that membralin co-localized with SYVN1, EMC1 and Derlin1, but not Ubxd8, VCP, or Ubiquilin2 (Supplementary Fig. 2e-j). Next, we determined whether membralin co-localizes with endogenous ERAD components in primary neurons. Immunostaining results confirmed co-localization of membralin with

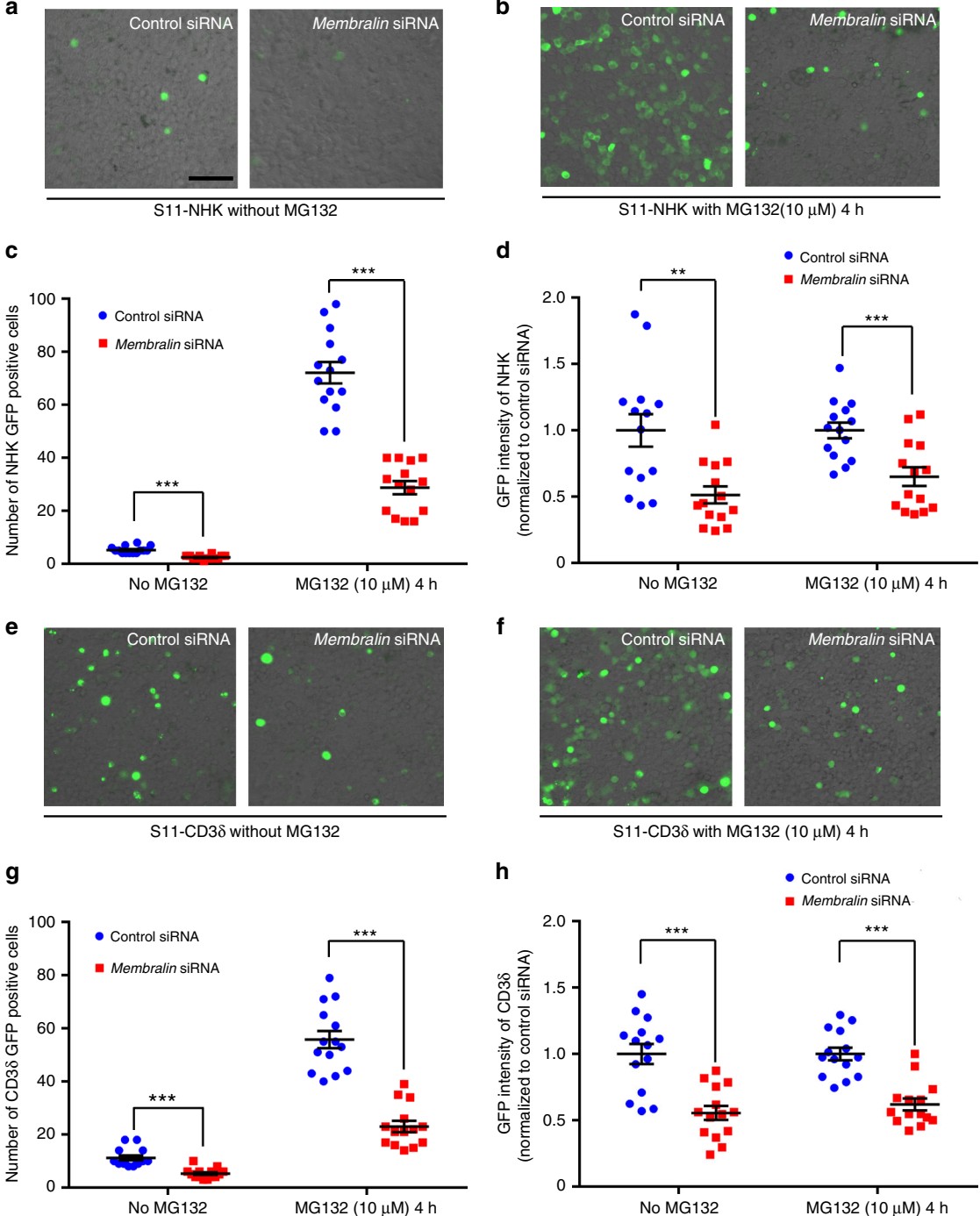

**Fig. 3** NHK (ERAD-L) and CD3δ (ERAD-M) degradation in HEK293T cells transfected with *membralin* siRNA. **a** Representative images depicting reconstituted NHK-GFP fluorescence were shown from cells transfected with control or *membralin* siRNA and subsequently co-transfected with S1-10 and SP-S11-NHK-HA without proteasome inhibitor MG132 treatment. **b** Representative images of NHK-GFP-positive cells transfected with control or *membralin* siRNA and exposed to MG132 (10 μM) for 4 h. **c** Quantification of NHK-GFP-positive cells with control and *membralin* siRNA transfection with or without MG132 treatment (*n* = 14 random field images per group). **d** Quantified GFP intensity in NHK-GFP-positive cells transfected with control or *membralin* siRNAs, with or without MG132 treatment (*n* = 14 random field images per group). **e** Representative images of reconstituted CD3δ-GFP fluorescence were shown from cells transfected with control or *membralin* siRNA and subsequently co-transfected with S1-10 and SP-S11-CD3δ-HA without proteasome inhibitor MG132 treatment. **f** Representative images of CD3δ-GFP-positive cells transfected with control or *membralin* siRNA and exposed to MG132 (10 μM) for 4 h. **g** Quantification of CD3δ-GFP-positive cells with control or *membralin* siRNA transfection with or without MG132 treatment (*n* = 14 random field images per group). **h** Quantified GFP intensity in CD3δ-GFP-positive cells transfected with control or *membralin* siRNAs, with or without MG132 treatment (*n* = 14 random field images per group). Data represent mean ± s.e.m. from three independent experiments. ***$P < 0.001$, unpaired *t*-test. Scale bar, 100 μm

SYVN1, AMFR and EMC3, with no co-localization observed with VCP (Supplementary Fig. 2k–n).

To confirm physical interactions between these protein components, we used fluorescence resonance energy transfer (FRET) to quantify membralin/protein interactions[26]. To this end, FRET efficiency was measured based on increases in GFP intensity following membralin-mCherry bleaching in

co-transfected donor/acceptor pairs. Representative pre- and post-bleaching images of SYVN1-GFP, AMFR-GFP and EMC3-GFP show an increased FRET signal following membralin-mCherry photobleaching (Fig. 2a). Representative images of other membralin-interacting proteins also illustrated similarly increased GFP FRET intensities after membralin-mCherry photobleaching (Supplementary Fig. 3a). Although membralin

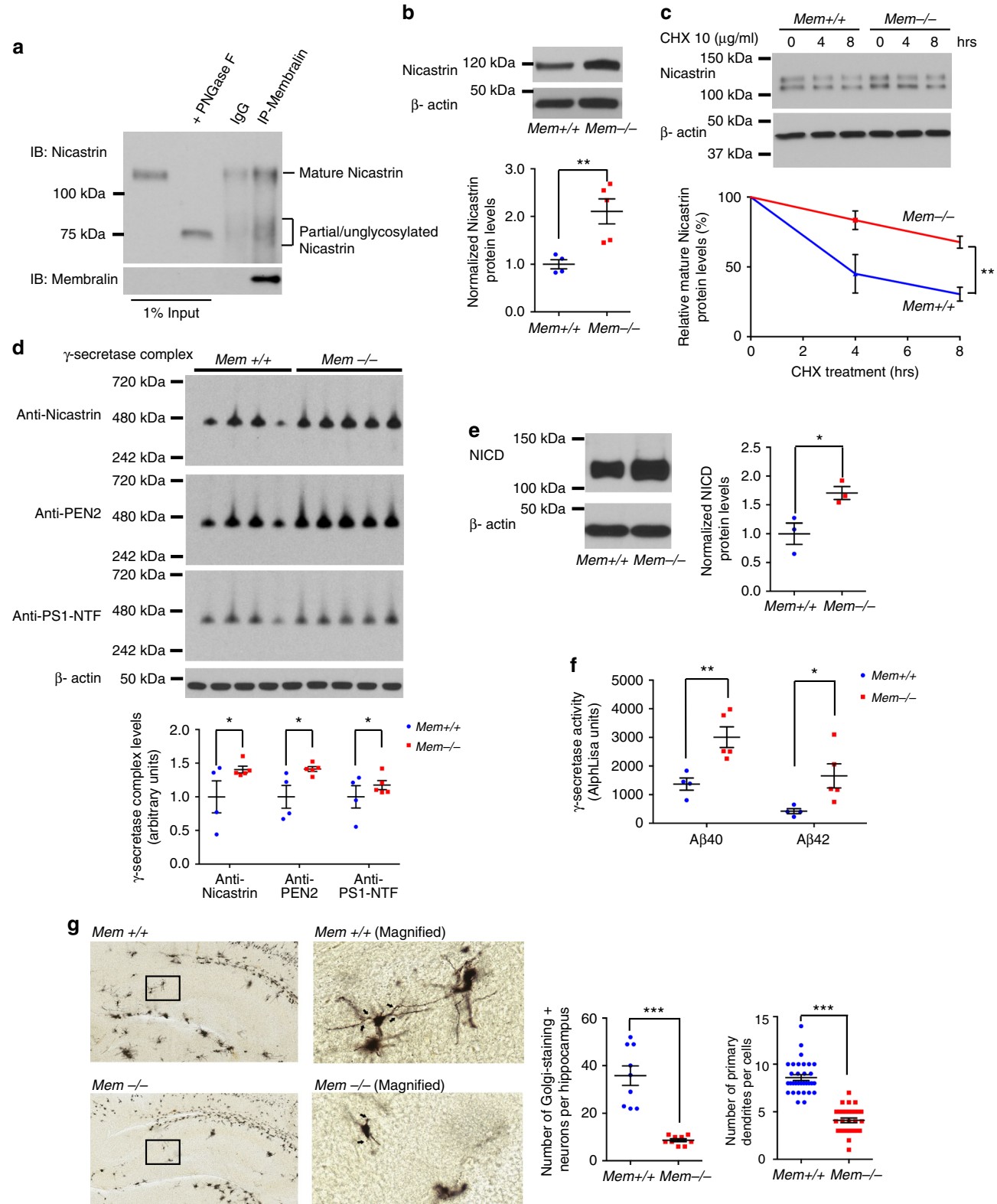

co-localized with EMC3, Ubac2, FAM8A1, AMFR, SYVN1, EMC1, and Derlin1 (Supplementary Fig. 2a–g and k–m), FRET efficiencies differed between membralin and the various interacting proteins identified (Fig. 2b): membralin interacted with Ubac2 and FAM8A1 (Supplementary Fig. 3a) at the highest FRET efficiencies measured at $33.14 \pm 2.22\%$ and $33.85 \pm 2.07\%$, respectively, followed by EMC3 (Fig. 2a) and EMC1 (Supplementary Fig. 3a) with FRET efficiencies at $26.30 \pm 1.14\%$ and $22.36 \pm 0.95\%$, the E3 ligases SYVN1 ($21.78 \pm 1.55\%$) and AMFR ($11.68 \pm 1.25\%$) (Fig. 2a), and Derlin1 at an efficiency of $12.30 \pm 1.07\%$ (Supplementary Fig. 3a). An EGFP-mCherry fusion protein joined by an alanine polylinker sequence (mCherry-7AA-EGFP) was used as a positive control[27], with a FRET efficiency measured at $44.95 \pm 2.26\%$ (Fig. 2b). No FRET was observed between membralin and Ubxd8 (Fig. 2b), which was consistent with our co-localization data (Supplementary Fig. 2h).

We further confirmed interactions between membralin and FRET-positive candidates by co-immunoprecipitation (co-IP). Myc-tagged membralin was transfected into HEK293T cells and the interacting complexes were co-precipitated using an anti-Myc antibody. We observed co-precipitation of endogenous SYVN1, AMFR, and EMC3 in Myc-membralin complexes (Supplementary Fig. 3b). Using antibodies to immunoprecipitate endogenous complexes, we found that endogenous membralin in N2a cells co-immunoprecipitated SYVN1 and AMFR (Fig. 2c). Further, we found that these interactions were enhanced under ER stress conditions induced by tunicamycin and thapsigargin (Fig. 2c and Supplementary Fig. 3c). These results further confirmed that membralin physically interacts with previously characterized ERAD components such as SYVN1 and AMFR, and together with our combined proteomic and FRET analysis, present strong evidence indicating that membralin is a part of the ERAD network.

**Membralin mediates both ERAD-L and ERAD-M degradation.** To further test the functional role for membralin in the ERAD complex, we employed a split-GFP-based protein dislocation assay to study the effects of membralin on the retro-translocation of ERAD substrates[28]. In this assay, a C-terminal GFP fragment (GFP strand 11, or S11) is fused with ER luminal α1-antitrypsin null Hong Kong variant (NHK), and CD3δ ER-membrane translocation substrates, respectively; and a complementary N-terminal GFP fragment (strands 1-10 of GFP, or S1-10) is expressed in the cytosol. Positive GFP signals are generated through retro-translocation of the S11 GFP-tagged ER substrates into the cytosol by the ERAD complex to associate with the S10 GFP reporter in the cytosol. Under steady-state conditions, we

found that membralin knockdown decreased both the number and the intensity of GFP-positive cells by nearly half using the ER-luminal NHK reporter system (Fig. 3a, c, d), and a similar reduction using the ER-membrane CD3δ reporter construct (Fig. 3e, g, h). These effects were enhanced in the presence of proteasome inhibitor MG132 (10 µM) which can inhibit cytosolic protein turnover (Fig. 3b, f). Transient expression of the substrate fragments was not altered by *membralin* siRNA downregulation (Supplementary Fig. 4a, b) and the knockdown efficiency of *membralin* siRNA was confirmed by quantitative PCR (Supplementary Fig. 4c). Together, our results indicate that membralin is an important ERAD component for the degradation of both ERAD associated luminal and membrane (ERAD-L and ERAD-M) substrates.

**Nicastrin is a membralin-dependent ERAD substrate.** Given that membralin is abundantly expressed in brain[20] and protein unfolding is significantly prominent in numerous neurodegenerative disorders[29,30], we specifically considered whether membralin could affect normal ER function and influence the degradation of protein components involved in neurodegenerative pathology. Previous studies shown that immature nicastrin can be trapped in the ER with PS1/PS2 deficiency[31–33]. Interestingly, we identified nicastrin as a potential binding component for membralin in our initial proteomic analysis of the membralin interactome (Fig. 1c, indicated in orange). We confirmed endogenous interactions between membralin and nicastrin: both mature (120 kDa) and unglycosylated (75 kDa) nicastrin co-immunoprecipitated with membralin from mouse brain organelle-enriched (nuclear- and cytosol-free) lysates (Fig. 4a). Co-localization of membralin and nicastrin was also examined in primary cortical neurons in culture by immunofluorescence (Supplementary Fig. 5a). Given that nicastrin is a key component of the γ-secretase complex, we further investigated the possibility that membralin may interact with other γ-secretase complex components, such as PS1 and PEN2. However, membralin immunoprecipitates from N2a membrane fractions show interactions with immature nicastrin (lower band), with no interaction detected with other γ-secretase complex components (Supplementary Fig. 5b–f). We next characterized potential membralin sub-regions that may interact with nicastrin. Since no definitive protein interaction domain modules have been identified in membralin, we designed several Myc-tagged constructs spanning major predicted transmembrane and cytosolic/ER-luminal loops (Supplementary Fig. 6a). All constructs were expressed in HEK293T cells except for N2 and C3 which expressed at low detection levels (Supplementary Fig. 6b). We found that both N1 and C2 constructs co-precipitated with

**Fig. 4** Nicastrin is a membralin-dependent ERAD substrate. **a** Endogenous co-IP of nicastrin with membralin. Mouse brain organelle-enriched (nuclear and cytosol-free) lysates were precipitated with an anti-membralin antibody and immunoblotted with anti-nicastrin and membralin antibodies as indicated. A lower molecular weight band (~ 75 kDa) in the nicastrin blot was the brain organelle-enriched lysates treated with PNGase F to remove the glycosylation of nicastrin, serving as unglycosylated control. Untreated sample was used for endogenous co-IP, and shown in the input. **b** Western blot analysis and quantification of nicastrin protein levels detected in neuronal tissue from *Mem +/+*($n = 4$ mice) and *Mem−/−* ($n = 5$ mice) animals. Data represent mean ± s.e.m. unpaired *t*-test, **$P < 0.01$. **c** Western blot analysis of nicastrin protein levels measured in *Mem+/+*or *Mem−/−* mouse NPCs treated with 10 µg/ml CHX for the indicated time. Data represent mean ± s.e.m. from three independent experiments, two-way ANOVA, **$P < 0.01$. **d** Native PAGE western blot analysis and quantification of γ-secretase subunits (nicastrin, PEN2 and PS1-NTF) in brains from *Mem+/+*($n = 4$ mice) and *Mem−/−* ($n = 5$ mice) animals normalized to β-actin levels from an equivalent quantity of denatured lysate. Data represent mean ± s.e.m. unpaired *t*-test, *$P < 0.05$. **e** Western blot analysis and quantification of NICD protein levels in *Mem+/+* and *Mem−/−* mouse NPCs. Data represent mean ± s.e.m. from 3 independent experiments, unpaired *t*-test, * $P < 0.05$. **f** γ-secretase activity was analyzed by measuring Aβ40 and Aβ42 generation in purified membrane fractions derived from *Mem+/+* ($n = 4$ mice) and *Mem−/−* ($n = 5$ mice) mouse brains. Data represent mean ± s.e.m. unpaired *t*-test, * $P < 0.05$, ** $P < 0.01$. **g** Golgi staining of the hippocampus of *Mem+/+* and *Mem−/−* mice (postnatal 3-day-old, $n = 3$ mice per genotype). Scale bars, 600 µm (left), and 60 µm in the magnified images (right). Number of Golgi-staining+ neurons per hippocampus ($n = 9$ sections per genotype) and number of primary dendrites per cell were quantified ($n = 30$-32 neurons per genotype). Data represent mean ± s.e.m. unpaired *t*-test, ***$P < 0.001$

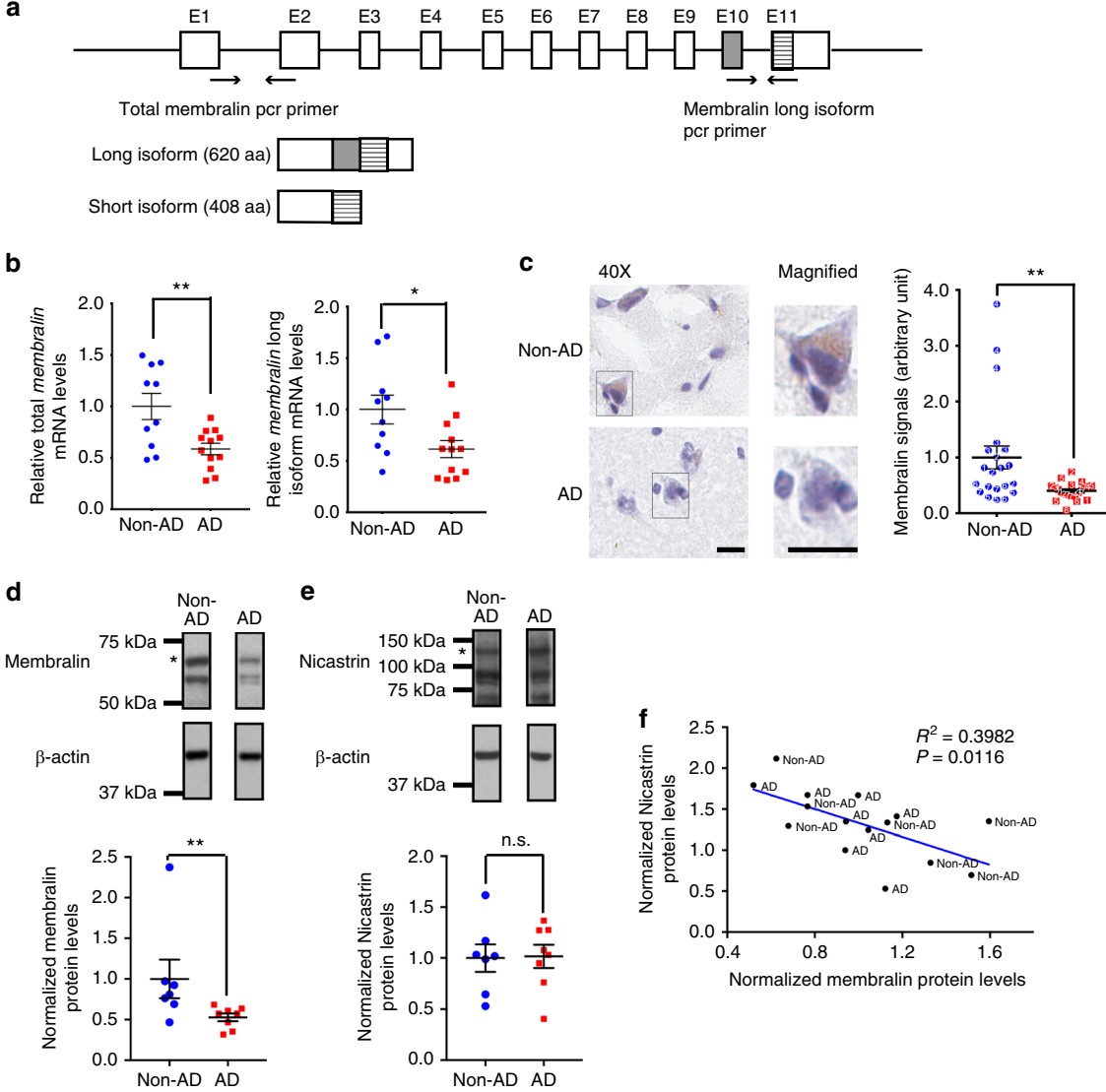

**Fig. 5** Membralin expression levels are dysregulated in AD brain. **a** Intron/exon distribution of human *membralin* long and short isoforms (top), and schematic depiction of *membralin* long (620aa) and short (408aa) protein isoforms (bottom). Primers used to detect total and *membralin* long isoforms were shown. **b** Quantification of total and long *membralin* transcripts in non-AD and AD brain normalized to the *18 s ribosomal RNA* control (*n* = 10 for non-AD samples; *n* = 12 for AD samples). Data represent mean ± s.e.m. unpaired *t*-test, * *P* < 0.05, **P* < 0.01. **c** Membralin expression detected by immunohistochemistry staining (*brown*) with hematoxylin counter-staining (*purple*) in human postmortem tissues (three random field images from each sample were quantified using Leica Aperio analysis software in a blinded manner from a total of *n* = 7 samples per group). Data generated from individual samples are indicated with a sample ID. Scale bars, 15 µm. Data represent mean ± s.e.m. unpaired *t*-test, **P* < 0.01. **d, e** Western blot analysis and quantification of **d** membralin and **e** nicastrin protein levels in non-AD (*n* = 7) and AD (*n* = 8) samples. Representative images from the same blots are shown. Asterisks indicate membralin **d** and nicastrin **e**. Data represent mean ± s.e.m. unpaired *t*-test, **P* < 0.01, n.s. no significance. **f** Correlation analysis of membralin (*X* axis) with nicastrin protein levels (*Y* axis) in non-AD and AD samples. Y = -0.3401 * X + 0.8727, $R^2$ = 0.3982, *P* = 0.0116 (*n* = 7 for non-AD samples, *n* = 8 for AD samples); the slope was found to be significantly deviant from non-zero by linear regression analysis

HA-nicastrin in the HEK293T cells (Supplementary Fig. 6c), which indicate that a 378-485aa membralin region containing the forth transmembrane domain was important for nicastrin interaction. Next, we investigated whether nicastrin could be a membralin-dependent ERAD substrate. Indeed, we observed a significant increase in nicastrin protein levels in membralin knockout (*Mem*−/−) neuronal tissues (Fig. 4b) and neural progenitor cells (NPCs) derived from *Mem*−/− mice relative to wild-type samples (*Mem* +/+) (Supplementary Fig. 7a); whereas no change in nicastrin mRNA levels was observed (Supplementary Fig. 7b). By performing a cycloheximide (CHX)-chase assay in wild-type and membralin deficient NPCs, we observed reduced nicastrin turnover rates in *Mem*−/− NPC samples in comparison to *Mem* +/+ NPCs (Fig. 4c). Together, these results indicate that nicastrin is a potential membralin-dependent ERAD substrate.

Since nicastrin is a key component of the γ-secretase complex, we determined whether membralin deletion affects γ-secretase complex integrity and function. Native PAGE analysis indicate the presence of a high-molecular-weight γ-secretase complex comprising the four key components (nicastrin, PEN2, APH-1 and PS1)[34,35], which significantly increased in membralin-deficient mouse brain samples by immunoblot using anti-nicastrin, anti-PEN2, and anti-PS1-NTF antibodies (Fig. 4d), suggesting that membralin deletion enhanced the integrity of the γ-secretase complex. Next, we determined whether membralin

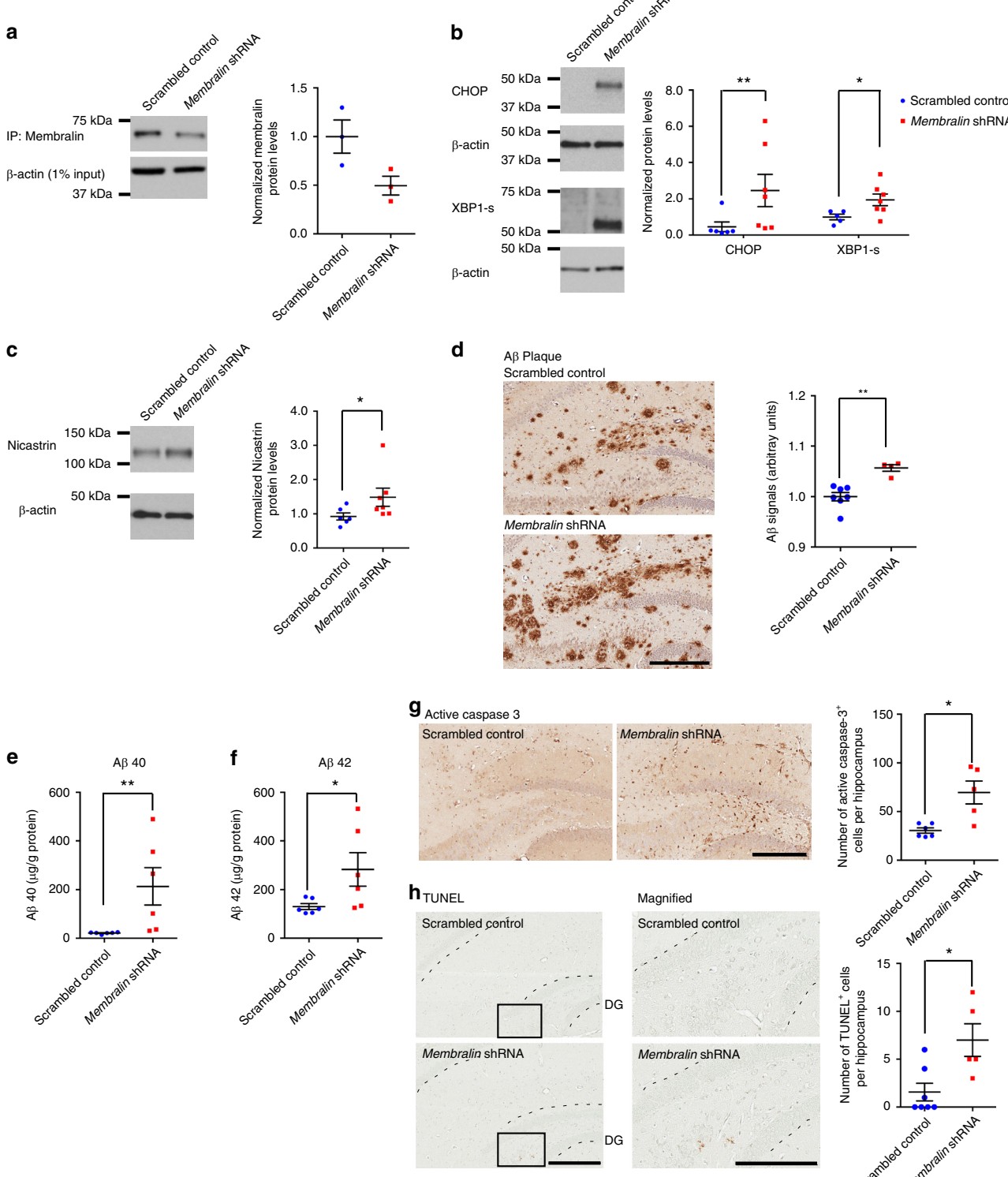

**Fig. 6** Modulation of membralin expression in TgCRND8 mice. **a** Membralin protein levels from TgCRND8 mouse hippocampus injected with scrambled control or *membralin* shRNA constructs. Protein levels were determined by immunoprecipitation and immunoblot using an anti-membralin antibody. Protein levels were normalized relative to β-actin (*n* = 3 mice per group). **b** XBP1-s and CHOP protein levels from TgCRND8 mouse hippocampus injected with scrambled control or *membralin* shRNA constructs. **c** Nicastrin protein levels normalized to β-actin in the hippocampus of scrambled control (*n* = 6) and *membralin* shRNA (*n* = 7) injected mice. **d** Aβ plaque staining and quantification was determined by staining with the MOAB2 Aβ antibody in the DG region (*n* = 7 for scrambled control group; *n* = 4 for *membralin* shRNA injected animals). **e** Human Aβ40 and Aβ42 levels in hippocampal lysates from 6-8-month-old TgCRND8 mice injected with scrambled control or *membralin* shRNA lentiviruses were determined by ELISA (*n* = 6 mice per group). **g**, **h** Histological analysis of 6-8-month-old TgCRND8 mice injected with scrambled control or *membralin* shRNA lentiviruses. Presence of active (cleaved) caspase 3 and apoptotic TUNEL staining in the dentate gyrus (DG) region and quantification is presented in **g** and **h**, respectively. Data represent mean ± s.e.m. unpaired *t*-test, *$P < 0.05$, **$P < 0.01$, and ***$P < 0.001$. Scale bars, 300 μm (left) and 200 μm in magnified images (right)

deletion could modulate downstream effects on the cleavage of γ-secretase substrates such as Notch and APP. We found that γ-secretase cleaved Notch NICD increased in membralin-deficient mouse NPCs (Fig. 4e). Similarly, γ-secretase dependent APP cleavage at both Aβ40 and Aβ42 sites[36,37] significantly increased in mouse membralin-deficient brain (Fig. 4f), suggesting that the activity of the γ-secretase complex was elevated with membralin deficiency. Given that homozygous membralin deletion is lethal at postnatal day 5, we examined the effects of membralin deletion on neuronal morphology *in vivo*. We observed a significant reduction in number of primary dendrites and Golgi-stained neurons in the hippocampus of postnatal 3-day-old *Mem−/−* mouse brains (Fig. 4g).

**Membralin mRNA and protein levels are dysregulated in AD**. As Aβ accumulation is an underlying event in AD, we questioned whether membralin dysfunction may be associated with AD. Previous reports showed the presence of two human membralin isoforms generated by alternative splicing[38], a long isoform comprising all 11 exons, and a short isoform lacking the 10th exon, leading to an early termination codon (Fig. 5a). Using quantitative PCR primers designed to identify total and long *membralin* transcripts (Fig. 5a), we characterized the expression of membralin in human AD brain. Both total *membralin* and long *membralin* transcripts were observed to decrease by 40% in AD (Fig. 5b). Further, we examined membralin expression in human postmortem brain tissue by immunostaining, and observed a marked reduction in membralin staining intensity in AD patient samples ($n = 7$) compared to the non-AD control ($n = 7$) (Fig. 5c). Likewise, we also observed decreased membralin expression in AD patient samples by immunoblot analysis (Fig. 5d), with no change in nicastrin expression levels observed in non-AD and AD patient samples (Fig. 5e). However, we did observe a significant inverse correlation between membralin and nicastrin levels in combined non-AD and AD patient samples (Fig. 5f, $R^2 = 0.3982$, $P = 0.00116$), which strongly suggests that membralin may modulate nicastrin levels and γ-secretase activity in aged brain and AD pathogenesis.

**Downregulating membralin levels can modulate Aβ pathology**. Since membralin can regulate γ-secretase activity through the modulation of nicastrin levels, we determined whether reducing membralin could aggravate Aβ-associated phenotypes in an AD mouse model. Lentiviral particles containing scrambled control or *membralin* shRNA were stereotactically injected into the dentate gyrus (DG) region of 6-8 month-old TgCRND8 mice[39]; membralin protein levels decreased by 50% in *membralin* shRNA animals compared to the control animals (Fig. 6a), confirming the efficacy of our shRNA knockdown system. To exclude potential off-target effects associated with *membralin* shRNA expression, we determined whether nicastrin protein levels could be rescued by co-expressing *membralin*-targeting shRNAs with shRNA-refractory *membralin* cDNA constructs. Nicastrin protein levels increased significantly in N2a cells transduced with *membralin* shRNAs, whereby co-expression of *membralin* shRNAs together with shRNA-refractory plasmid restored nicastrin protein levels comparably to control levels (Supplementary Fig. 8a, b). Elevated CHOP and XBP1-s (XBP-1 spliced form) were observed in *membralin* shRNA-injected hippocampus (Fig. 6b), indicating aberrant activation of the UPR system induced upon membralin depletion. Consistent with our results with membralin deletion (Fig. 4b) and human postmortem samples (Fig. 5f), we observed a significant increase in nicastrin levels in *membralin* shRNA

samples (Fig. 6c) and an increased Aβ plaque load in the *membralin* shRNA animals (Fig. 6d), which was accompanied by a concomitant increase in human Aβ40 and Aβ42 levels (Fig. 6e, f). Given cytotoxic effects associated with Aβ, we examined shRNA-injected animals for AD-like pathology. We observed a significant increase in apoptotic cell death in *membralin* shRNA-injected animals as indicated by increases in active (cleaved) caspase-3 staining in the DG region compared to the control animals (Fig. 6g). Similarly, TUNEL-positive cells were also found in *membralin* shRNA animals, with little staining observed in control animals (Fig. 6h).

Furthermore, we also detected reductions in PSD95 levels with *membralin* shRNA expression (Fig. 7a), indicating potential synaptic defects that may be associated with membralin down-regulation. In agreement with this notion, we found that membralin knockdown increased the primary escape latency compared to the control animals, denoting a defect in memory function as determined using Barnes maze tests (Fig. 7b, c). Similarly, in contextual fear conditioning tests, mice subjected to membralin knockdown showed significant reductions in freezing time after electrical foot-shock compared to control animals (Fig. 7d, e), indicating impairments in memory with membralin downregulation. Together, our results demonstrate that membralin-associated ERAD defects can aggravate Aβ accumulation to enhance neuronal death, synaptic dysfunction, and memory deficits in an AD mouse model.

## Discussion

Protein folding and homeostasis are essential to physiological brain function and development. Thus, protein quality control mechanisms such as the ERAD system are essential to monitor proper protein folding and to direct misfolded targets for degradation[6–8]. Although components of the ERAD system have been recently characterized through proteomic analysis[14], it remains unclear whether certain key components within the ERAD machinery have yet to be identified. Our results here that implicate membralin as an essential component of the ERAD system provide insight into how the ERAD complex functions. Further, these results define a previously unknown role for membralin in maintaining proper homeostatic turnover of pathogenic substrates such as nicastrin, which may define alternative strategies to attenuate pathogenic onset in neurodegenerative disorders such as AD.

Previous reports have indicated a role for the ERAD system in maintaining proper function of various neuronal receptors and ER stress response systems. For example, cell surface GABA_B receptor levels, and IRE1α, the essential mediator of the unfolded protein response system, are two substrates which are regulated by ERAD degradation[40,41]. Moreover, ERAD components have been shown to mediate the degradation of inositol trisphosphate (IP_3) receptors[42] and immune cell surface components such as major histocompatibility complex (MHC) class I molecules[43], implicating additional roles for ERAD in membrane signal transduction and antigen presentation. Given our results that membralin can mediate translocation of both luminal and membrane ER substrates, it is likely that membralin function coincides with a variety of cellular ERAD substrates. Indeed, other potential targets of interest in the membralin interactome such as TMEM66 may also be potential membralin-dependent ERAD substrates. The role of membralin in the degradation of additional ERAD substrates warrants future investigation.

Protein aggregation and proteopathic formation of Aβ-enriched plaques and phospho-tau-containing NFTs are hallmarks of AD which appear at disease onset[44,45]. As an age-associated

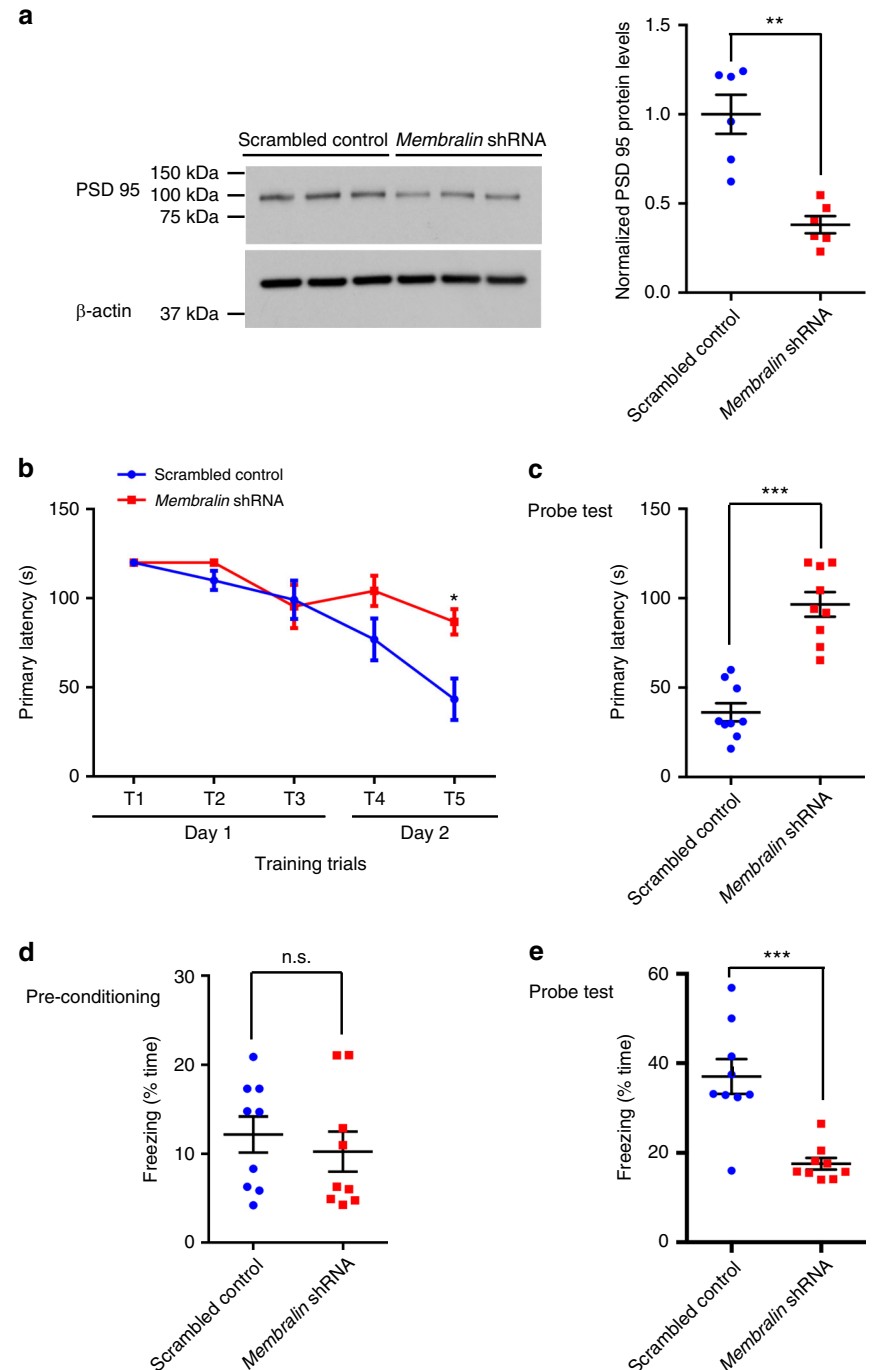

**Fig. 7** AD-associated behaviors in TgCRND8 mice injected with membralin shRNA. **a** PSD95 levels in scrambled control or *membralin* shRNA-injected hippocampal tissue as determined by immunoblot, and normalized relative to β-actin ($n = 6$ mice per group). **b**, **c** Barnes maze analysis: **b** Comparison of primary latency to target in control or *membralin* shRNA-injected groups in five training trials ($n = 9$ mice per group). **c** Comparison of primary latency to target in control or *membralin* shRNA-injected groups in probe tests ($n = 9$ mice per group). **d**, **e** Fear conditioning analysis: **d** Comparison of freezing time (% time) in shRNA-injected experimental cohorts ($n = 9$ mice per group) during the pre-conditioning trial. **e** Comparison of freezing time (% time) following electrical foot-shock in shRNA-injected experimental cohorts ($n = 9$ mice per group) during the probe test. Data represent mean ± s.e.m. unpaired *t*-test, *$P < 0.05$, **$P < 0.01$, and ***$P < 0.001$

disorder, AD is associated with prolonged neuronal exposure to proteotoxic and oxidative stressors which have been observed to induce ER stress and activation of UPR machinery in neurons from AD brain[21]. Although neurotoxic elements which compromise proper protein folding accumulate in aged AD brain, a clear connection between AD and other ER protein monitoring systems such as ERAD have yet to be firmly established. Previous

studies indicate that membralin may be alternatively spliced in AD[25], and an SNP in proximity to the *membralin* gene locus is linked to late-onset AD[24]. Our findings here show that membralin is an essential component of the ERAD complex which is downregulated in AD. Importantly, we identified a significant inverse correlation between membralin levels and nicastrin expression in human postmortem brain samples. Given that our

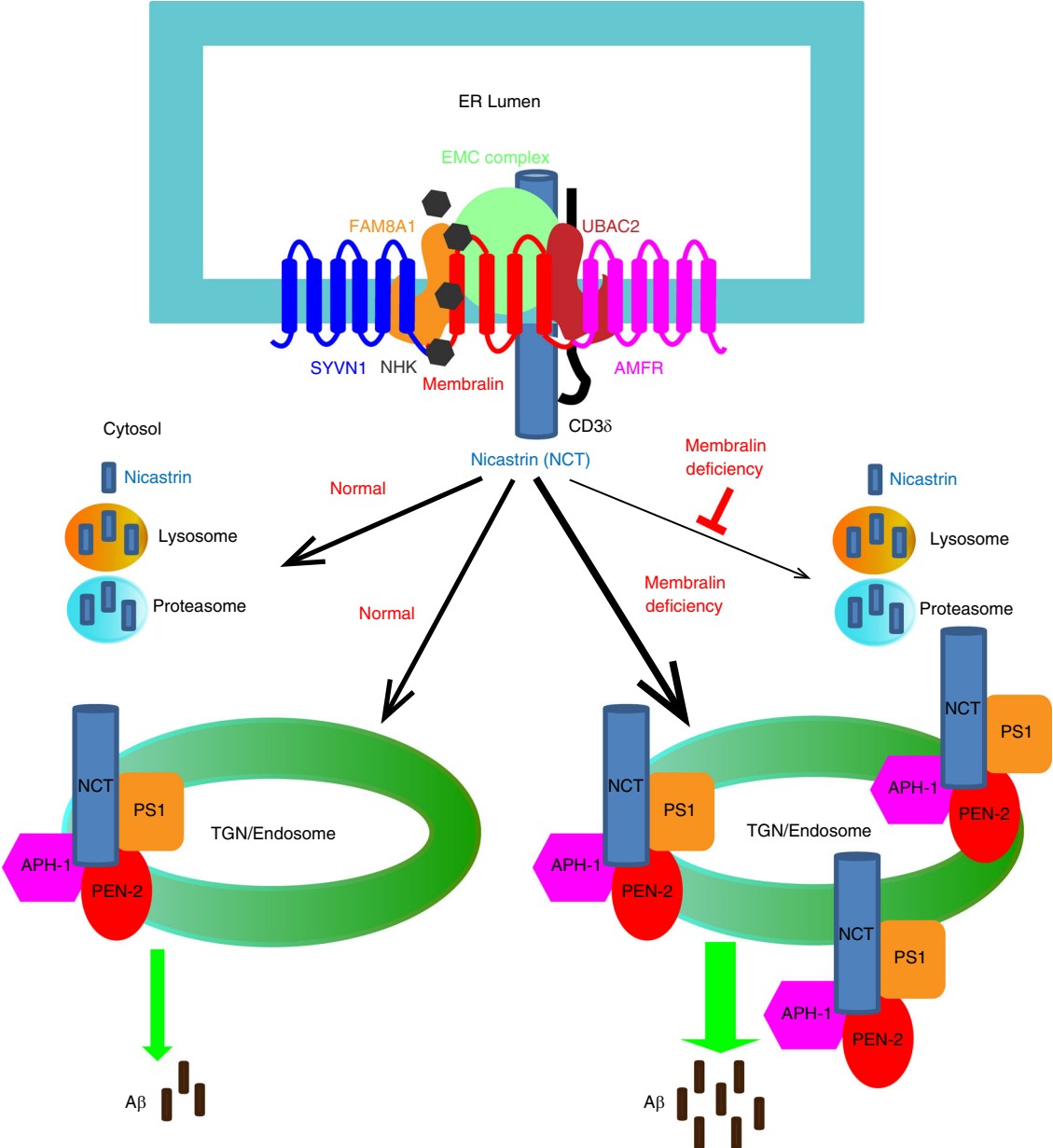

**Fig. 8** A schematic model of the ERAD component membralin in regulating nicastrin and Aβ generation. Membralin interacts with several known ERAD components and mediates nicastrin degradation. Membralin deficiency increases nicastrin levels through inhibiting nicastrin degradation, thereby increasing Aβ generation

results indicate that membralin levels can influence nicastrin expression and consequent γ-secretase function, factors that dysregulate or attenuate membralin may contribute to the pathological events such as Aβ accumulation, plaque deposition and memory impairment in AD (Fig. 8).

Although it is not yet known whether membralin is down-regulated in most AD cases, other components within the membralin interactome have been implicated with potential roles linked to AD onset. For example, the membralin-interacting component Erlin2 was also found to interact with nicastrin[46], while the membralin-associated E3 ubiquitin ligase SYVN1 (Hrd1) was previously observed to degrade the γ-secretase inhibitor Rer1[47]. These findings together with our results described here, corroboratively indicate that Aβ generation can be altered by perturbations in the ERAD system. This raises the possibility that restoring ERAD function may be an alternative means to attenuate Aβ accumulation and toxicity in AD.

Our focus here characterizing membralin-associated ERAD nicastrin degradation has led us to establish a role for membralin in Aβ-associated AD pathology and cognitive decline. Previously, we have reported that the membralin homozygous deletion in mice results in postnatal lethality at day 5, where no obvious phenotypic changes are apparent in heterozygous membralin deficient animals[20]. Heterozygous membralin deficient animals (age > 6 month-old) also show no apparent cognitive differences compared to wildtype littermates in Barnes maze and contextual fear conditioning tests (Supplementary Fig. 9). A slight but statistically non-significant elevation in nicastrin levels was observed in heterozygous membralin deficient animals compared to wild-type animals (Supplementary Fig. 7c). This suggests that membralin from a single copy may be sufficient to sustain normal physiological ERAD function in the absence of age-related proteotoxic stress. However, in the presence of elevated Aβ levels, adult brain may be more sensitized to changes in membralin/

ERAD function, thereby revealing cryptic cognitive and behavioral deficiencies with partial reductions in membralin levels. In addition, our observation that membralin can affect γ-secretase dependent Notch cleavage and activation suggests that ERAD modulation may also alter Notch-associated developmental pathways. We previously described γ-secretase dependent Notch cleavage as a developmental trigger for enlarged brain ventricles in hydrocephalus in sorting nexin 27 (SNX27) deficient mice[35]. It remains possible that γ-secretase-induced hyperactivation of the Notch signaling pathway may perturb critical developmental events contributing to early postnatal lethality with homozygous membralin deletion. Together, with our results here, membralin-associated regulation could potentially influence numerous developmental and neuropathological pathways through γ-secretase substrates such as Notch, and APP as demonstrated here, and may therefore be a key player in maintaining proper protein homeostasis in a variety of physiological functions.

In summary, our results identify membralin as an ERAD component and demonstrate a role for membralin-associated ERAD function in AD. This may provide further insight into potential therapeutic targets for AD, and other neurological disorders.

## Methods

**Antibodies**. The mouse-specific membralin antibody used in this study was previously custom-made and now is available from Millipore (ABN1661, 1:1000). The PS1-NTF antibody used in this study was described previously (Rabbit polyclonal antibody, 1:1000)[35]. Other antibodies were acquired from commercial sources as follows: c-Myc (Sigma, C2956, 1:1000), HA (Cell Signaling Technology, 3724 s, 1:1000), EMC3 (Santa Cruz, sc-365903, 1:1000), SYVN1 (Sigma, H7790, 1:500), SYVN1 (Santa Cruz biotechnology, sc-293484, 1:1000), AMFR (Cell Signaling Technology, 9590 s, 1:1000), AMFR (Santa Cruz Biotechnology, sc-166358, 1:1000), VCP (Thermo Fisher Scientific, MA3-004, 1:1000), VCP (Cell Signaling Technology, 2648 s, 1:1000), PS1-CTF (Millipore, 5232, 1:2000), human membralin (Sigma, HPA042669, 1:500), nicastrin (Cell Signaling Technology, 9447 s, 1:1000), nicastrin (Millipore, MAB5556, 1:1000), β-actin (Sigma, 5441, 1:10000), NICD (Cell Signaling Technology, 4147 s, 1:1000), PEN2 (Cell Signaling Technology, 8598 s, 1:1000), PSD95 (Cell Signaling Technology, 3450 s, 1:1000), active (cleaved) caspase3 (Cell Signaling Technology, 9662 s, 1:100), MOAB2 (Aβ40/42, Biosensis, M-1586-100, 1:500), rabbit IgG control (Cell Signaling Technology, 2729, 1 μg used as IP control).

**Animals**. Membralin knockout (Mem−/−) C57BL/6 mice were generated using a gene-trapping method previously described[20]. TgCRND8 mice in a hybrid C3H/He-C57BL/6 background were provided by Dr. Xiongwei Zhu from Case Western Reserve University and Dr. Peter St. George-Hyslop from University of Toronto. All mice were age and sex-matched and the sample size were estimated before experiments. Animals are randomly assigned to experimental groups and no animals were excluded from the analysis. The investigators performing experimental procedures and sample processing were blinded to genotype within the duration of the experiments/data acquisition. Both membralin knockout mice and TgCRND8 mice used here were housed in reverse day/night cycle and given access to food and water. Experimental procedures performed approved and supervised by Sanford Burnham Prebys Medical Research Institute (SBP, La Jolla, CA) Institutional Animal Care and Use Committee. Female mice were housed with littermates in groups of two to three and male mice were housed only one per cage. The same female and male mice were used for both Barnes maze and fear conditioning behavior tests during day cycles.

**Cell culture and transfection**. Primary cortical neurons were obtained by microdissection of the cerebral cortex from E17-E18 embryos (C57BL/6 mice) using a stereomicroscope, and dispersed by digestion in trypsin and DNaseI for 30 min. at 37 ℃ followed by trituration in DMEM. Neurons were plated and maintained on poly-L-lysine coated coverslips in Neurobasal medium supplemented with B27, glutamine and penicillin/streptomycin, whereby media was changed every 3 days, where half of the media was replaced.

Mouse NPCs were isolated from E13.5 embryos dissected from Mem+/+ and Mem−/− mice. NPCs were cultured in suspension with NPC proliferation medium (NSC basal medium + NSC proliferation supplement (STEMCELL Technologies) + 20 ng/ml EGF + 10 ng/ml bFGF). The NPCs between passages 4-6 were used in experimental procedures.

N2a cells were a gift from Dr. Gopal Thinkaran (University of Chicago), and maintained in DMEM supplemented with 10% FBS. Plasmid constructs were transfected into N2a cells using Genjet™ in vitro DNA Transfection Reagent (SignaGen@ Laboratories) according to the manufacturer's specifications (1 μg/well, 24-well plates).

HEK293T cells were a gift from Dr. William Stallcup (SBP), and maintained in DMEM supplemented with 10% FBS. siRNAs were transfected at a concentration of 10 nM using Lipofectamine RNAiMAX transfection reagent (Thermo Fisher Scientific) according to the manufacturer's specifications. Control siRNA (#4390843) and membralin siRNA duplexes (S40619 and S40620) were purchased from Thermo Fisher Scientific. Plasmid constructs were transfected using Attractene transfection reagent (Qiagen) according to the manufacturer's specifications (1.2 μg per dish for 35 mm dishes and 0.4 μg per well for 24-well plates).

**Plasmid construction**. cDNAs encoding mouse membralin, SYVN1, AMFR, EMC3, EMC1, Ubac2, Derlin1, VCP, Ubxd8, Ubiquilin2 were purchased from the Integrated Molecular Analysis of Genomes and their Expression (IMAGE) consortium and subcloned into pEGFP-N1, pmCherry-N1 or pcDNA3.1-Myc vectors using standard molecular biological methods.

The plasmids pCMV-mGFP(1–10) and pCMV-mGFP(Cterm S11) were purchased from Theranos-tech, Inc. To construct plasmids expressing the ER-localized S11-NHK-HA and S11-CD3δ-HA, DNA fragments encoding an N-terminal signal peptide (the first 33 amino acids from NHK) followed by mGFP S11, a linker sequence (GDGGSGGGSAS), the NHK (aa26-357) or mouse CD3δ (aa20-173), and a C-terminal HA tag sequence were inserted into the EcoRI/NotI sites of pCMVmGFP(Cterm S11) for NHK or into the EcoRI site of pCIneo for CD3δ.

A shRNA-refractory membralin overexpression construct was designed to overexpress mouse membralin with concurrent membralin shRNA expression targeting the following sequence: 5′-ctcatcaacgttcgcgatcggctattcca-3′. Forward 5′-ctcatcaacgttcgcgatcggctattcca-3′ and reverse 5′-tggaatagccgatcgcgaacgttgatgag-3′ primer sequences were used to integrate silent mutations into human and mouse cDNAs (pCDNA3.1 membralin-myc) comprising 3 nucleotide mismatches in the original shRNA targeting sequence.

All subcloned plasmid constructs were confirmed by sequencing.

**Membralin rescue assay**. N2a cells were transduced with scrambled control or membralin shRNA viruses at multiplicity of infection (MOI) 10 with TransDux Max reagent (System Biosciences); 48 h following transfection, shRNA-refractory membralin plasmids were transfected into N2a cells and cultured for an additional 2 days. Proteins from cell lysates were collected at day 4 for immunoblot analysis.

**Sample preparation for proteomic analysis**. HEK293T cells were transfected with plasmids expressing Myc-tagged mouse membralin or an empty Myc vector. 3 days after transfection, three independent HEK293T samples were extracted with Digitonin, Triton X-100 and NP40 IP lysis buffer (Thermo Fisher Scientific), respectively. The samples were then immunoprecipitated with anti-Myc magnetic beads overnight. Following immunoprecipitation, proteins were digested using Trypsin/Lys-C mix (Promega): briefly, IP and control samples were resuspended in 8 M urea per 50 mM ammonium bicarbonate after washing with 50 mM ammonium biocarbonate. Cysteine alkylation was achieved with 30 mM iodoacetamide (IAA), sheltered from light at room temperature for 30 min after reducing disulfide bonds in 10 mM tris(2-carboxyethyl) phosphine (TCEP) at 30 ℃ for 60 min. Urea (1 M) was prepared in 50 mM ammonium bicarbonate. Sample digestion was performed in MS grade Trypsin/Lys-C mix overnight. Magnetic beads were used to precipitate the digested peptides the next day, followed by washing once in 50 mM ammonium bicarbonate to recover the peptides. Finally, the digested samples were desalted using a C18 TopTip (PolyLC) column and dried in a SpeedVac system.

**Liquid chromatography (LC)-MS/MS analysis**. Peptide samples were prepared for LC-MS/MS analysis by reconstitution with 2% acetonitrile/0.1% formic acid, and injected into a 0.180 × 20 mm C$_{18}$ trap symmetry column (Waters Corp.) coupled with an analytical C$_{18}$ BEH130 PicoChip column 0.075 × 100 mm, 1.7 μm particles (NewObjective) mounted on a nanoACQUITY Ultra Performance Liquid Chromatography system (Waters Corp.). Peptides were separated with a 210-min linear gradient of 2-28% solvent B at a flow rate of 400 nL/min. The column outlet was directly coupled to an Orbitrap Velos Pro MS (Thermo Fisher Scientific) operated in positive data-dependent acquisition mode. MS1 spectra were measured with a resolution of 60,000, an AGC target of $10^6$ and a mass range from 350 to 1400 m/z. Up to 5 MS2 spectra per duty cycle were triggered, fragmented by collision-induced dissociation, and acquired in the ion trap with an AGC target of $10^4$, an isolation window of 2.0 m/z and a normalized collision energy of 35.

**Proteomic data analysis**. All mass spectra data were analyzed using MaxQuant software, version 1.5.2.8. Briefly, MS/MS spectra were searched against the Human Uniprot protein sequence database in addition to the mouse membralin protein sequence (all versions April 2015). Precursor mass tolerance was set to 20 ppm and 4.5 ppm for the initial search where mass recalibration was completed and for the

main search, respectively. Product ions were analyzed with a mass tolerance of 0.5 Da. The maximum precursor ion charge state used for the searching was set at 7. Carbamidomethylation of cysteines was searched as a fixed modification, while oxidation of methionines was probed as a variable modification. Enzyme was set to trypsin in a semi-specific mode and a maximum of two missed cleavages was allowed in the search. The target-decoy-based false discovery rate (FDR) filter for spectrum and protein identification was set to 1%. Second peptide mode of MaxQuant software was also enabled. Identified proteins were queried against the Crapome database comprised of 411 experiments. A heatmap was generated using the R Bioconductor gplots package.

**Co-localization and FRET assays**. HEK293T cells were transfected for 24 h with a pair of target-EGFP and membralin-mCherry fusion proteins. The fluorescence intensities of EGFP and mCherry in transfected cells were examined using a Zeiss 710 NLO microscope (Carl Zeiss Inc.). For co-localization experiments, images were acquired using a 63× oil lens. For FRET experiments, a 40× water lens was used to acquire three pre-bleached and five post-bleached images. Averaged fluorescence intensities of the donor before and after bleaching were calculated from the measurement of 10 cells in three independent experiments per group. The efficiency of FRET was calculated by $E_{fret} = 1 - (I_a/I_b)$, where $I_a$ and $I_b$ represents the steady-state donor fluorescence in the presence and absence of the acceptor, respectively. An EGFP-mCherry fusion protein linked by seven alanines (mCherry-7AA-EGFP) was used as the FRET-positive control.

**Co-immunoprecipitation**. HEK293T cells were transfected with a Myc-membralin construct for 3 days. Following transfection, cells were trypsinized and extracted in IP buffer (Thermo Fisher Scientific). Total protein concentration was measured by bicinchoninic acid (BCA) assay (Thermo Fisher Scientific) according to the manufacturer's instructions. Protein lysate (1 mg) was incubated with 1 µg of c-Myc antibody or 1 µg rabbit IgG negative control, together with 50 µl protein A/G magnetic beads (Thermo Fisher Scientific) at 4 °C overnight. One percent of the cell lysate was collected as an input sample. The protein–antibody–bead complex was washed with IP wash buffer (Thermo Fisher Scientific) three times and subjected to elution by boiling in 1× SDS loading buffer. The interaction between endogenous membralin with ERAD components was assayed in N2a whole cell homogenates in 1% digitonin lysis buffer. The interaction between endogenous membralin and nicastrin was assayed in extracts derived from organelle-enriched fractions generated from mouse brain samples. Mouse brain samples were homogenized in a 1× isotonic EB buffer (10 mM Hepes, pH7.8, 250 mM sucrose, 25 mM KCl, and 1 mM EGTA) supplemented with Complete protease inhibitor cocktail (Roche), followed by sonication and two rounds of serial centrifugation to remove nuclear and cytosol fractions: (1) 1000 x g for 15 min, where the supernatant was subsequently spun; (2) 140,000 × g for 2 h. Pellets were lysed in a lysis buffer (50 mM Tris-HCl, pH8.0, 150 mM NaCl, 1% NP40), supplemented with the Complete protease inhibitor cocktail (Roche). Protein lysates (500 µg) were incubated with membralin antibody and Protein G magnetic beads (Thermo Fisher Scientific) at 4 °C overnight. The protein–antibody-bead complex was washed with IP wash buffer three times and subjected to elution by boiling in 2× SDS loading buffer. Input controls represent 1% of the total pellet lysates. A 1% pellet lysate volume was also treated with PNGase F (Promega) and served as an unglycosylated control.

**Protein substrate dislocation assay**. HEK293T cells were first transfected with 10 nM either control or *membralin* siRNA for 48 h using Lipofectamine RNAiMAX (Thermo Fisher Scientific), and subsequently co-transfected with S1-10 and SP-S11-NHK-HA plasmids or S1-10 and SP-S11-CD3δ-HA plasmids for 24 h. MG132 (10 µM) was added to relevant samples for 4 h, and GFP-positive cell images were acquired using a Zeiss 710 NLO microscope.

**Cycloheximide chase assay**. Mouse NPC cells were treated with CHX (10 µg/ml) for 4 and 8 h, respectively. At the end of each treatment, cells were lysed in radioimmunoprecipitation assay (RIPA) lysis buffer and subjected to SDS-PAGE and immunoblotting.

**Hippocampal stereotactic injection**. Control and mouse-targeting *membralin* shRNA constructs were packaged into lentiviral particles where 2 µl at a titer of 5 × 10^8 TU/ml were stereotactically injected into the DG of TgCRND8 mice at 6–8 months of age at the following coordinates: anterior posterior, −2.0 mm; medial lateral, ± 1.3 mm; dorsal ventral, 2.1 mm[48]. Lentiviral particles were packaged by the Viral Core at Sanford Burnham Prebys Medical Discovery Institute. To confirm region-specific membralin knock-down in the DG region, mice were anesthetized and sacrificed 5 weeks after injection, where brain tissues were rapidly dissected and processed. Hippocampal lysates were prepared by homogenizing tissue in RIPA buffer in the presence of protease inhibitors for analysis by immunoblotting.

**Mouse behavioral analysis**. Barnes maze test was performed as described previously[49]. Briefly, TgCRND8 mice were habituated on day 1, where mice were

placed in the center of the maze underneath a clear 3,500-ml glass beaker for 30 s. Mice were then slowly guided to the target hole leading to the escape cage by moving the glass beaker over the target hole within a span of 10–15 s. The mice were then given 3 min to enter the target hole, and were gently forced to enter in the event no entry was apparent. Training was initiated on day 2. Mice were placed inside an opaque cardboard cylinder, 10″ tall and 7″ in diameter, in the center of the Barnes maze for 15 s. The cylinder was then removed and mice were allowed to explore maze for 2 mins. If the mouse entered the target hole/escape cage within this time, it was allowed to remain in the escape cage for 1 min; otherwise the mouse was gently guided to the escape hole using the glass beaker and allowed to freely enter the escape cage. In the case where the mouse did not enter the escape cage within 3 min, it was nudged with the beaker to gently force entry. Five trails was used for training, 3 trials on training day 1, and 2 trials on training day 2. Forty-eight hours after the last training session, a probe test to analyze the behavioral characteristics of the mice seeking the target area was monitored using a digital camera using the ANY-maze video tracking system (Stoelting Co.). Subsequent analyses of the probe test parameters were processed using ANY-maze software, where statistical analyses and significance values were calculated using GraphPad Prism. Contextual fear conditioning behavior test was performed using the Freeze Detector System (San Diego Instruments, CA). Twenty-four hours before the training day, the mice were placed in the conditioning chamber and allowed to freely explore the surrounding for 5 mins. On the training day, the mice were placed in the conditioning chamber, and the mice were allowed to freely explore the chamber for 120 s, where a 0.4 mA electrical foot-shock was subsequently applied to the mice for 2 s. After 60 s, another 0.4 mA shock was given to the mice for 2 s. Following shocks, the mice were left in the chamber for an additional 120 s. Twenty-four hours after training, each mouse was monitored in the same chamber for 5 min. Freezing time was automatically recorded and analyzed by the Freeze Detector System.

**Postmortem Alzheimer's disease brain samples**. Postmortem human Alzheimer's disease brain samples used (Supplementary Table 2 and 3) in this study were provided by Dr. Eliezer Masliah from University of California, San Diego, CA and University of Miami (Neurobiobank) with informed consent from the donors, and were analyzed under ethical and safety guidelines approved by the SBP Institutional Review Board (IRB) and under California and National Institutes of Health guidelines.

**Immunoblotting**. Mouse brain samples and cell samples were lysed in the RIPA buffer, supplemented with Halt^{TM} Protease and Phosphatase Inhibitor Cocktail (Thermo Fisher Scientific), and followed by sonication and centrifugation. Equal protein quantities derived from cell or tissue lysates were subjected to gel electrophoresis (4–12% Bolt Bis-Tris gel, Thermo Fisher Scientific) and probed using antibodies as indicated. Images have been cropped for presentation. Full size images are presented in Supplementary Fig. 10.

**Immunohistochemistry**. Immunohistochemical analysis was performed as described previously[50,51]. Briefly, mice were anesthetized and fixed by intracardial perfusion with PBS followed by 4% paraformaldehyde (PFA). Brain tissues were harvested and fixed in a 4% PFA solution and subsequently embedded in paraffin for sectioning. Coronal brain sections (5 µm) were deparaffinized, hydrated, and immunostained using antibodies as indicated, followed by diaminobenzidine (DAB) staining using a DAB peroxidase substrate kit (Vector Laboratories). TUNEL staining was performed using the ApopTaq® Peroxidase In Situ Apoptosis Detection kit (Milipore, S7100). Golgi staining was performed using the FD Rapid GolgiStain Kit (FD NeuroTechonologies). All the images were acquired using a ScanScope® AT2 system. Active caspase 3^+ and TUNEL^+ cells were quantified using NIH ImageJ software in a blinded manner. Human membralin and Aβ plaque positive staining was quantified using Leica Aperio analysis software in a blinded manner. Non-AD human samples and scrambled control shRNA-injected animals were used to normalize relevant datasets to 1.0. Golgi-stained images for dendrite quantification were acquired using a Zeiss 710 NLO microscope under differential interference contrast (DIC) with a 40× water lens and analyzed using NIH ImageJ software.

**Aβ ELISA assay**. Mouse hippocampi were isolated and lysed in RIPA buffer, supplemented with protease inhibitors (Roche). Human Aβ40 and Aβ42 levels were quantified using ELISA detection systems (Life Technologies) according to the manufacturer's instructions. Briefly, standards and diluted samples (50 µl) were added to the appropriate wells. Human Aβ 40/ Aβ 42 antibody solutions (50 µl) were added into each well besides the chromogen blank well. The plates were covered and incubated overnight at 4 °C. The next day, anti-rabbit IgG HRP solution (100 µl) were added into each well except the chromogen blank well and the plate was incubated for 30 min at room temperature. Stabilized chromogen (100 µl) were added into all the wells. The plate was incubated in the dark at room temperature for 15 min. The absorbance of each well was read at 450 nm after adding the stop solution.

**Statistics**. All results were repeated at least three times in independent experiments. Data were presented as mean ± s.e.m. and statistical significance was determined using unpaired $t$-test, one way ANOVA, or two-way ANOVA. The variance is similar between the groups that are being statistically compared. Data sets were analyzed for statistical significance using GraphPad Prism software (version 7). A $P$ value < 0.05 was considered as statistically significant.

**Data availability**. The data that support the findings of this study are available from the corresponding author upon reasonable request.

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

## Acknowledgements

We dedicate this article to the memory of Dr. Dongxian Zhang. This work was supported by "Laboratory Funding Initiative" from Sanford Burnham Prebys Medical Discovery Institute and by grants from National Institute of Health (R01 HD074961 to D.Z.; R21

AG048519, R01 AG021173, R01 AG038710, R01 AG044420, and R01 NS046673 to H.X.) as well as from the Tanz Family Fund (to H.X.) and the Cure Alzheimer's Fund (to H.X.).

## Author contributions

B.Z., L.J, D.Z. and H.X. designed the experiments. B.Z., L.J., T.H., Y.Z., T.L. and X.L. performed experiments and analyzed the data. Y.Z. and E.M. provided key experimental materials. A.C. and K.P. performed proteomic experiments and analyzed the data. All authors discussed the results. B.Z., L.J., T.H., D.Z. and H.X. wrote the manuscript.

## Additional information

**Competing interests:** The authors declare no competing financial interests.

