## [Peer Review File · Nature Communications]

Reviewers' comments:

Reviewer #1 (Remarks to the Author):

This manuscript by Zhu et al. is an interesting manuscript linking the ERAD protein membralin to the secretase complex and processing of APP. Though the first part of the paper on the association of membralin with ERAD is strong, the link between membralin and APP biology needs further support. The major points and controls raised below should strengthen the role of membralin in AD and the conclusions of this paper.

- The overexpression of ER proteins can lead to artificial localization. The authors should assess co-localization of Membralin and the ERAD components shown in Sup Fig 2 using antibodies to stain endogenous proteins.
- The same comment for the IP experiment in Figure 2c. The authors should immunoprecipitate endogenous membralin. Additionally, this IP experiments will benefit from a negative control e.g. ubiquitin 2. It does look like only a small percentage of the endogenous ERAD components interact with the overexpressed membralin. Does stimulating the ERAD pathway alter the association between membralin and the ERAD components?
- The results presented in Figure 3 are interesting. The authors should make sure the expression levels of the reporter constructs (S11-NHK and S11-CD3d) does not change between the non-silencing and siRNA conditions.
- Does knockdown of membralin (or knockout in neurons) alter ERAD components or degradation of ERAD substrates in general, or is the effect specific to nicastrin?
- Does Membralin interact with PS1 or PEN2?
- Figure 4d requires a loading control. As such the increased nicastrin and PEN2 levels are not convincing.
- Is there Abeta plaque pathology (or Abeta increase) in the membralin KO mice (in the wild-type animals without APP Tg)?
- Is the expression data in Figure 5b normalized to a loading control e.g. GAPDH RNA?
- How is the protein expression data in Figure 5c quantified? Are the cells chosen to be analyzed in a blind-manner?
- These data can be strengthened if the authors provide biochemical data showing the reduction in membralin levels in AD tissues.
- The data shown in Figure 6 needs to be strengthened. In Figure 6a, a reduction in protein

levels of membralin is not clear in the shRNA lanes - if anything the protein levels seem to go up with the shRNA. Is the labeling correct in Figure 6a? The plaque staining in Figure 6b and 6f needs to be quantified. It is not clear if the effects of membralin shRNA is due to knockdown of membralin itself, or due to an off-target effect. The authors can either use independent shRNAs to show similar phenotypes or rescue the defects with an RNAi-insensitive cDNA. Behavior data in Figure 1h and i are not convincing - the authors need to show the training data for these behavior tests to rule out pre-existing differences in the ability to perform these tasks. Additionally, wt non-transgenic animals should be shown as controls.

Minor comments:

In Figure 1B, what does the intensity color scale represent? peptide #, peptide AUC etc.

In Figure 4A, it is not clear what the input is for the IP. Is it the PNGaseF treated sample or the untreated sample.

Reviewer #2 (Remarks to the Author):

The manuscript by Zhu et al is focused on the analysis of ER protein membralin and its potential connection with AD. Membralin is ER protein with several predicted transmembrane domains. Its function in cells is poorly understood. GWAS study pointed to an AD-associated SNP locus in close proximity of membralin gene. However, mechanistic connection between membralin and AD has not been established.

In a series of affinity purification mass spectrometry experiments the authors identified membralin-associated proteins. Bioinformatics analysis lead to hypothesis that membralin plays a role in ER-associated degradation system (ERAD). Association of membralin with ERAD components was confirmed by co-localization and FRET analysis. Its role in ERAD was further established in retro-translocation assay of ERAD substrates. Interestingly, nicastrin was identified as membralin binding partner and as membralin-dependent ERAD substrate. These findings provided a link to AD, as reduction of membralin levels is shown by the authors to result in upregulation of gamma-secretase complex and enhanced production of Aβ.

Overall, these are very interesting, innovative and important findings that for the first time provide mechanistic link between membralin and AD. I have just a few concerns/suggestions.

1. Fig 2B and Sup Fig 3B. Please combine these panels on Fig 2B so efficiency of membralin association with different ERAD components can be compared side-by-side.

2. Fig 2C. IP data for EMC3 and SYVN1 are not convincing. Association with AMFR is strong in IP assay but relatively weak in FRET assay. This is confusing. My suggestion is to remove IP data from the paper unless better quality data can be obtained

3. Fig 3 – ERAD assay. The main observed effect of membralin knockdown is increased number of GFP positive cells. Was the intensity of GFP signal quantified? It is expected that changes in ERAD degradation of split GFP substrates should also affect intensity of the GFP signal.

4. Fig 4C – Nicastrin turnover assay. Starting amounts of nicastrin are very different in Mem^{+/+} and Mem^{-/-} cells. It may effect the measurements of turnover rate. Nicastrin levels are more comparable on Fig 4B.

5. Fig 4G – reduction in a number of neurons and primary dendrites in the brains of 3 day old Mem^{-/-} mice. These effects are likely to be related directly to the loss of membralin function and not mediated by Abeta. Abeta typically does not cause neuronal and synaptic loss that quickly. Also, these mice do not express human APP gene.

6. Fig 6A – levels of membralin appear to be elevated in shRNA lines, not reduced. Probably this gel is mislabeled.

Reviewer #3 (Remarks to the Author):

In this ms, Zhu and colleagues address an important question of endoplasmic-reticulum associated degradation (ERAD) in neurodegeneration. They provided studies to suggest that membralin is a component of ERAD. Interestingly, they found that this ER protein is decreased in brains of AD patients and provide data to suggest that nicastrin, a component of gamma-secretase complex, is membralin associated ERAD substrate. In addition, they showed that reduction in membralin led to exacerbation of amyloid pathology and memory deficits in TgCRND8 mice. While these data of are interest, several issues as indicated below need to be addressed in order to substantiate conclusions drawn in ms.

1. To exclude over-expression effects of transfection studies used in this ms, it will be important to show endogenous interactions of membralin with other components of ERAD in cultured cells.

2. Fig 5: While the authors showed decreased levels of membralin in AD patients, is the level of Nct increased in these AD patients? What about other components of the gamma-secretase?

3. Fig 6: It is not clear as to why IP-Blot showed membralin level higher in shRNA against membralin when compared to control shRNA (Fig. 6a); is Nct level increased with membralin shRNA treatment? It will strengthen this ms if there is evidence to exclude the possibility of off-target effects of membralin shRNA.

4. Since previous work showed that membralin +/- mice have no reported phenotype (Yang et al, eLife 4, 2015), it is surprising that 50% reduction of membralin is sufficient to show phenotype using membralin shRNA in TgCRND8 mice (Fig 6). To resolve this apparent discrepancy or potential off-target effects of shRNA, it will be critical to access effects of caspases, Nct levels, etc. in membralin +/- mice.

5. Fig 6: To substantiate their model that ERAD is impacted when membralin is down

regulated, it will be important to show that ERAD defects occurs in AD mice treated with shRNA against membralin.

Reviewers' comments:

Reviewer #1 (Remarks to the Author):

This manuscript by Zhu et al. is an interesting manuscript linking the ERAD protein membralin to the secretase complex and processing of APP. Though the first part of the paper on the association of membralin with ERAD is strong, the link between membralin and APP biology needs further support. The major points and controls raised below should strengthen the role of membralin in AD and the conclusions of this paper.

We thank the reviewer for these comments. We have indeed further characterized interactions between membralin and its ERAD components, and interactions with other gamma-secretase components. Importantly, we also demonstrate a negative correlation between membralin levels and nicastrin levels in human brain samples, as well as a significant reduction in membralin levels in human AD.

• The overexpression of ER proteins can lead to artificial localization. The authors should assess co-localization of Membralin and the ERAD components shown in Sup Fig 2 using antibodies to stain endogenous proteins.

We have determined whether membralin co-localized with endogenous ERAD components in primary neurons. In our revised submission, we now show endogenous co-staining of membralin with SYVN1, AMFR, EMC3 and VCP in cultured neurons in the revised Supplementary Figure 2; we see some co-localizing overlap with all components except for VCP.

• The same comment for the IP experiment in Figure 2c. The authors should immunoprecipitate endogenous membralin. Additionally, this IP experiments will benefit from a negative control e.g. ubiquitin 2. It does look like only a small percentage of the endogenous ERAD components interact with the overexpressed membralin. Does stimulating the ERAD pathway alter the association between membralin and the ERAD components?

We have now made substantial additions to revised Fig. 2c. We now show co-immunoprecipitation of endogenous membralin with SYVN1 and AMFR, and demonstrate that ER-stress induced by thapsigargin and tunicamycin enhanced endogenous membralin interaction with SYVN1 and AMFR.

With respect to a negative co-IP control, we don't have an ubiquitin antibody appropriate for co-immunoprecipitation. However, we have performed endogenous co-IP experiments with VCP/membralin and detected no interaction (shown below). As this is a negative result, we prefer not to include this panel.

- **The results presented in Figure 3 are interesting. The authors should make sure the expression levels of the reporter constructs (S11-NHK and S11-CD3d) does not change between the non-silencing and siRNA conditions.**

We have now included quantified data in Supplementary Figure 4 to show that membralin siRNA transfection did not affect S11-NHK-HA or S11-CD3 δ -HA expression, and had no effects on GFP (S1-10) expression.

- **Does knockdown of membralin (or knockout in neurons) alter ERAD components or degradation of ERAD substrates in general, or is the effect specific to nicastrin?**

We believe that membralin downregulation may alter the degradation of other ERAD substrates, and the effects we observe may not be specific to nicastrin. We are indeed interested in exploring this further, and we have mentioned/described this possibility in the revised Discussion.

- **Does Membralin interact with PS1 or PEN2?**

We did not detect PS1 or PEN2 in our proteomic analysis. We have also attempted to detect membralin interactions with PS1/PEN2 by co-IP, and failed to detect interactions with these components. Further, we found that membralin interacts with immature nicastrin, which is primarily accumulated in the ER. This may explain why membralin only interacts with nicastrin, but not other γ -secretase components. We present these results in our revised Supplementary Fig. 5b-f.

- **Figure 4d requires a loading control. As such the increased nicastrin and PEN2 levels are not convincing.**

We have exerted substantial effort to improve the quality of this figure, and reassessed the abundance of a high molecular weight γ -secretase complex by native PAGE analysis. New panels in the revised Fig. 4d now show a clear increase in nicastrin, PEN2, and PS1 (NTF) in a high molecular weight complex with membralin deletion using non-denatured brain samples. The same amount of protein lysates were immunoblotted under denaturing conditions and probed for β -actin as a loading control.

- **Is there Abeta plaque pathology (or Abeta increase) in the membralin KO mice (in the wild-type animals without APP Tg)?**

It has been well established that murine A β is less prone to aggregation and proteotoxicity, and therefore mouse models will not form A β plaques in the absence of human APP or A β . Thus, both membralin WT and knockout mice will not form A β plaques in the absence of a human APP transgene.

We also note that homozygous membralin deletion is lethal at postnatal day 5, and will therefore be unable to establish membralin KO/APP transgenic lines required to perform a comparative analysis of A β plaque pathology in WT and membralin KO mouse lines.

- **Is the expression data in Figure 5b normalized to a loading control e.g. GAPDH RNA?**

The expression data was normalized to *18S rRNA* as a loading control.

- **How is the protein expression data in Figure 5c quantified? Are the cells chosen to be analyzed in a blind-manner?**

The positive staining was scored using Leica Aperio analysis software, and random fields were analyzed in a blinded manner. A more detailed description of the analysis methods used are included in the revised manuscript (Methods section).

- **These data can be strengthened if the authors provide biochemical data showing the reduction in membralin levels in AD tissues.**

We show in the revised submission that membralin levels are decreased in AD patients compared to controls in Fig. 5d. We have additionally performed a correlation analysis between membralin and nicastrin levels in human brain, and observe an inverse correlation between membralin expression and nicastrin levels. This further supports our conclusion that membralin dysfunction or deficiency can upregulate nicastrin levels and γ -secretase levels in human neuropathogenesis.

- **The data shown in Figure 6 needs to be strengthened. In Figure 6a, a reduction in protein levels of membralin is not clear in the shRNA lanes - if anything the protein levels seem to go up with the shRNA. Is the labeling correct in Figure 6a? The plaque staining in Figure 6b and 6f needs to be quantified. It is not clear if the effects of membralin shRNA is due to knockdown of membralin itself, or due to an off-target effect. The authors can either use independent shRNAs to show similar phenotypes or rescue the defects with an RNAi-insensitive cDNA. Behavior data in Figure 1h and i are not convincing - the authors need to show the training data for these behavior tests to rule out pre-existing differences in the ability to perform these tasks. Additionally, wt non-transgenic animals should be shown as controls.**

We thank the reviewer for pointing out the original mislabeling of Fig. 6a in the control and membralin shRNA samples. This was an image processing error, and has been fixed in the revised figures. Original blots which were correctly labeled are available for confirmation upon request.

We have now added the quantitation of A β plaques to our revised Fig. 6d, and TUNEL staining to Fig. 6h.

To confirm that our results using shRNA expression were not derived from off-target artifacts, we have co-expressed membralin shRNAs and refractory membralin cDNA constructs, and determined consequent effects on nicastrin levels in Supplementary Figure 8. We observe that membralin shRNA elevated nicastrin levels, which were reduced to levels observed with scrambled shRNA controls with membralin shRNA/refractory cDNA co-expression. These results confirm that results from membralin shRNA expression were not due to off-target effects.

We have now included training data in our revised Fig. 7b for Barnes Maze experiments, and Fig. 7d for fear conditioning experiments. We did not see any differences between control and *membralin* shRNA-injected animals prior within early stages of training (Barnes Maze), or during pre-conditioning (contextual fear conditioning).

Please note that due to space limitations, the original Fig. 6g-i together with the behavioral analysis results (training data) have been combined to the new Fig. 7 in this revised submission.

Minor comments:

In Figure 1B, what does the intensity color scale represent? peptide #, peptide AUC etc.

The color intensity shown in Fig. 1b represents protein intensity (summed peptide area under the curve, or AUC). We have now included a description in the revised figure legend.

In Figure 4A, it is not clear what the input is for the IP. Is it the PNGaseF treated sample or the untreated sample.

The IP input for Fig. 4a is untreated; we have clarified this in the revised submission.

Reviewer #2 (Remarks to the Author):

1. Fig 2B and Sup Fig 3B. Please combine these panels on Fig 2B so efficiency of membralin association with different ERAD components can be compared side-by-side.

Due to the number of image panels presented in these figures, we have left the image panels separated as in our original submission. However, we have merged the quantified FRET analysis of all the ERAD components into a single graph in Fig. 2,b which will effectively provide a side-by-side comparison of relative interactions between these components by FRET.

2. Fig 2C. IP data for EMC3 and SYVN1 are not convincing. Association with AMFR is strong in IP assay but relatively weak in FRET assay. This is confusing. My suggestion is to remove IP data from the paper unless better quality data can be obtained

As suggested by the editor, we have improved this figure rather than removing it from our revised submission. We have therefore replaced the data originally presented in Fig. 2c with panels of higher quality; indeed endogenous membralin interactions with ERAD components such as AMFR are relatively weak at steady state. These interactions are seen to increase with ER-stress (as suggested in comments from Reviewer 1 above).

3. Fig 3 – ERAD assay. The main observed effect of membralin knockdown is increased number of GFP positive cells. Was the intensity of GFP signal quantified? It is expected that changes in ERAD degradation of split GFP substrates should also affect intensity of the GFP signal.

We have included quantification of GFP intensity measurements from these experiments in the revised Fig. 3d and h.

4. Fig 4C – Nicastrin turnover assay. Starting amounts of nicastrin are very different in Mem+/+ and Mem-/- cells. It may affect the measurements of turnover rate. Nicastrin levels are more comparable on Fig 4B.

As membralin deletion results in steady-state increases in nicastrin by approximately 2-fold (as seen in Fig. 4b), nicastrin levels will be consistently higher in membralin KO samples at the 0 timepoint. However, we have now included panels in revised Fig. 4c that may reflect more comparable nicastrin levels in Fig. 4b.

5. Fig 4G – reduction in a number of neurons and primary dendrites in the brains of 3 day old Mem^{-/-} mice. These effects are likely to be related directly to the loss of membralin function and not mediated by Abeta. Abeta typically does not cause neuronal and synaptic loss that quickly. Also, these mice do not express human APP gene.

We agree with the reviewer that these early effects seen in membralin KO pups are not due to effects on A β . As the activation of Notch pathway has been shown to exert effects on neuronal development, and given that Notch is a γ -secretase substrate, the effects seen may be due to Notch cleavage and activation. We mention/discuss this possibility in the revised Discussion.

6. Fig 6A – levels of membralin appear to be elevated in shRNA lines, not reduced. Probably this gel is mislabeled.

As mentioned above (Reviewer 1), we apologize for the mislabeling of this figure, which was derived from an image processing error. This has been corrected, and the original blots are available for verification.

Reviewer #3 (Remarks to the Author):

1. To exclude over-expression effects of transfection studies used in this ms, it will be important to show endogenous interactions of membralin with other components of ERAD in cultured cells.

We now include co-localization of membralin with ERAD components in Supplementary Figure 2, and include endogenous co-IP results in revised Fig. 2c (SYVN1) and Fig. 2d (AMFR). We note that endogenous membralin interactions as assayed by co-IP are enhanced with ER stress.

2. Fig 5: While the authors showed decreased levels of membralin in AD patients, is the level of Nct increased in these AD patients? What about other components of the gamma-secretase?

Changes in levels of components in the γ -secretase complex are not typically seen in AD, and in support of this, we also do not detect any change in nicastrin levels in AD patients compared to controls. However, we have performed a correlation analysis in human brain tissue, and demonstrate a negative correlation exists between membralin expression and nicastrin levels in human brain in revised Fig. 5f.

3. Fig 6: It is not clear as to why IP-Blot showed membralin level higher in shRNA against membralin when compared to control shRNA (Fig. 6a); is Nct level increased with membralin shRNA treatment? It will strengthen this ms if there is evidence to exclude the possibility of off-target effects of membralin shRNA.

We apologize as Fig. 6a control and shRNA panels were switched due to an image processing error. The error has been fixed, and the original blots bearing correct labels are available for verification. NCT levels increased with membralin shRNA treatment, which is shown in Fig. 6c.

We now include in Supplementary Figure 8, evidence that membralin shRNA increases nicastrin levels, and co-expression of membralin shRNA in combination with an shRNA-refractory membralin construct can abrogate this effect to exclude the possibility of off-target effects.

4. Since previous work showed that membralin +/- mice have no reported phenotype (Yang et al, eLife 4, 2015), it is surprising that 50% reduction of membralin is sufficient to show phenotype using membralin shRNA in TgCRND8 mice (Fig 6). To resolve this apparent discrepancy or potential off-target effects of shRNA, it will be critical to access effects of caspases, Nct levels, etc. in membralin +/- mice.

Importantly, we note that characterization of membralin +/- heterozygous phenotypes described previously were performed using cultured MEFs, where our study described here has been determined *in vivo* in combination with an AD mouse background (human APP with AD-associated Swedish/Indiana mutations). Additional AD-associated proteotoxicity may provide an explanation to the appearance of phenotypes seen in membralin +/- animals described in our study.

In agreement with this notion, we have performed Barnes maze and contextual fear conditioning tests comparing WT and membralin +/- animals, and have observed no difference in behavior between these two genotypes (in the absence of a mutant APP transgene). Since these experiments produced negative results, the data has not been included in our revised submission. However, we have compared nicastrin levels in WT, membralin +/- and homozygous membralin -/- knockout mice, and find that nicastrin levels are only significantly upregulated with homozygous membralin deletion. The results have been added to Supplementary Figure 7c, and described in the revised text.

5. Fig 6: To substantiate their model that ERAD is impacted when membralin is down regulated, it will be important to show that ERAD defects occurs in AD mice treated with shRNA against membralin.

Indeed, we now show in revised Fig. 6b that membralin shRNA triggers the expression of ER stress components CHOP and XBP1-s. Upregulation of these factors are indicative of upregulated unfolded protein response (UPR); we have included these results and an appropriate description in the revised Figure 6b and text.

Again, we appreciate all three reviewers for their extremely valuable and constructive comments, which as we have addressed has greatly improved the quality of our manuscript.

REVIEWERS' COMMENTS:

Reviewer #1 (Remarks to the Author):

"The authors did additional experiments and controls to strengthen the biochemistry, and imaging experiments. The animal model data in Figure 6-7 is also substantially improved. The manuscript is ready for publication in Nature Communications."

Reviewer #2 (Remarks to the Author):

The revised manuscript has been significantly improved. Addition of endogenous co-localization and IP data (Fig 2c) supports association of membralin with ERAD components. It is also important the the authors demonstrate negative correlation between membralin and nicastrin levels in human brains (Fig 5f). Other concerns regarding quality of the data in the paper has been addressed. I think the paper is acceptable in the present form.

Reviewer #3 (Remarks to the Author):

In their revised ms, the authors have addressed my previous concerns satisfactory.